# Easier to Judge than to Find: Predicting In-Context Learning Success for Demonstration Selection

Haochun Wang[1]   Chaofen Yang[1]   Jiatong Liu[1]   Jingbo Wang[1]   Zewen Qiang[1]   Sendong Zhao[1]
Bing Qin[1]   Ting Liu[1]

## Abstract

In-context learning (ICL) is highly sensitive to which demonstrations appear in the prompt, but selecting them is expensive because the space of possible demonstration contexts and combinations is enormous. We argue that demonstration selection is *easier to judge than to find*: predicting whether a specific query–context pair $(q, D)$ will succeed is cheaper and more general than searching for an optimal $D^\star$. Based on this insight, we propose DiSP, a sample-and-judge framework that stratifies queries by difficulty. DiSP runs random demonstration trials to estimate success rate of each training query, trains a lightweight router to predict difficulty from the query, and trains level-specific judges for sampled demonstrations. At inference, DiSP performs stop-on-acceptance judging under an explicit budget, emitting diagnostic risk tags when no suitable context is found. Across five classification datasets with Llama 3–8B and Qwen 2.5–7B, DiSP achieves the best average accuracy, improving over strong learned selection baselines by up to 3.4%, while achieving up to 23× end-to-end wall-clock speedup.

## 1. Introduction

Large language models (LLMs) can solve new tasks by conditioning on a few labeled input–output examples in the prompt, a phenomenon known as *in-context learning* (ICL) (Brown et al., 2020). Despite its practical impact, ICL is notoriously sensitive to demonstrations: which examples are chosen and how they are ordered can change the prediction from correct to confidently wrong (Lu et al., 2022). This sensitivity makes demonstration selection inherently

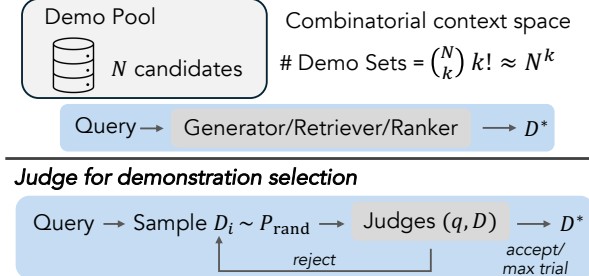

*Figure 1.* Finding vs. judging for demonstration selection. Searching for an optimal $D^\star$ faces a combinatorial space, while judging enables efficient sample-and-test with stop-on-acceptance under an explicit budget.

combinatorial. Given a candidate pool of size $N$, selecting $k$ demonstrations and ordering them yields $\binom{N}{k}k!$ possible demonstrations. Brute-force evaluation is typically infeasible because each candidate demonstration requires a full LLM inference, while real deployments often cannot afford running the LLM over many candidate demonstrations.

A common approach treats demonstration selection as an instance-adaptive *search* problem: map a query $q$ to a high-performing demonstration $D^\star$ via heuristic retrieval, learned retrieval/ranking, or proxy scoring with model feedback (Rubin et al., 2022; Iter et al., 2023; Nguyen & Wong, 2023; Ji et al., 2025; Zhang et al., 2025). Recent prompt optimization systems further treat demonstrations (and instructions) as tunable components in larger pipelines (Khattab et al., 2023; Opsahl-Ong et al., 2024). These methods can be effective, but they highlight a practical tension. Reliable search would ideally validate many candidates by running the LLM on $K$ candidate demonstrations, yet these additional LLM calls consume the very resource we aim to save. Moreover, proxy signals are imperfect—even semantically reasonable demonstrations can still cause failures and the same demonstration can behave differently across queries.

We propose an alternative framing: demonstration selection is *easier to judge than to find*. Instead of directly searching for $D^\star$, we learn a success predictor that answers a simpler question: *given a specific pair $(q, D)$, will ICL succeed?*

[1]Research Center for Social Computing and Interactive Robotics, Harbin Institute of Technology, China. Correspondence to: Sendong Zhao <sdzhao@ir.hit.edu.cn>.

*Proceedings of the 43rd International Conference on Machine Learning*, Seoul, South Korea. PMLR 306, 2026. Copyright 2026 by the author(s).

For a frozen LLM $M$, define the success indicator

$$s(q, D) = \mathbb{I}[M(q; D) \text{ is correct}], \qquad (1)$$

and learn a predictor $g(q, D) \approx P(s(q, D) = 1 \mid q, D)$ that is orders of magnitude cheaper to judge than running $M$. This judge enables a *sample-and-judge* strategy: sample candidate demonstrations from a fixed proposal distribution, evaluate them with $g$, then query $M$ with an accepted demonstration.

Building on this view, we introduce DiSP (**Di**fficulty-Stratified **S**uccess **P**rediction), a judge-centric framework that allocates compute for context selection only when it is likely to pay off. Unlike search-centric pipelines that train task-specific retrievers/rankers, DiSP keeps $M$ frozen and trains only compact discriminative modules. At inference, computation is a bounded number of cheap judge evaluations, yielding substantial wall-clock savings in test-time demonstration selection.

Crucially, the value of spending compute on demonstration selection is highly query dependent. Under a fixed proposal distribution over demonstration, some queries succeed for most sampled contexts, some are brittle and succeed only for a small fraction, and others are unlikely to be solved within any practical sampling budget. DiSP makes this heterogeneity explicit by estimating the success rate of each training query under random demonstration trials and stratifying queries into four difficulty levels ($l_1/l_2/l_3/l_x$), corresponding to distinct compute regimes under the fixed proposal.

Given these strata, we train (i) a lightweight router $r(q)$ that predicts the difficulty level from the query alone and (ii) level-specific judges that vet $(q, D)$ pairs. Judge capacity increases from $l_1$ to $l_3$ so that expensive features are reserved for brittle queries. At inference, the router selects a per-level sampling budget and judge to run stop-on-acceptance feasibility testing over random demonstrations; if $\hat{\ell} = l_x$, we skip judging, query the LLM once with a random demonstration, and emit HARD_QUERY. If no demonstration is accepted within budget, we fall back to a random demonstration and emit a NO_GOOD_DEMO tag. The sampling budgets provide an explicit accuracy–cost knob, while the risk tags expose when random-context ICL is likely to be unreliable. Empirically, on five classification benchmarks with two open-source backbones, DiSP improves average accuracy over strong selection baselines while reducing end-to-end wall-clock cost by up to $23\times$.

Our contributions are as follows:

- **A judge-centric sample-and-judge framework for demonstration selection.** We argue that predicting ICL success for a given $(q, D)$ is easier than searching for $D^\star$, and introduce DiSP with a router, level-specific

judges, and stop-on-acceptance feasibility testing that provides explicit cost knobs and diagnostic risk tags.

- **Random-trial supervision and query stratification.** We use offline random demonstration trials with the target LLM to estimate per-query success rates and define four difficulty levels that supervise routing and level-specific judging.

- **Efficiency and adaptability.** Across five classification benchmarks with Llama 3–8B and Qwen 2.5–7B, DiSP improves over strong learned selection baselines by up to 3.4% while achieving up to $23\times$ end-to-end wall-clock speedup.

## 2. Related Work

### 2.1. Mechanisms of In-Context Learning

Existing work studies how pre-trained Transformers perform new tasks from a few examples, including explanations based on implicit optimization (Von Oswald et al., 2023; Shen et al., 2024; Mittal et al.), implicit Bayesian inference (Xie et al., 2022), and mechanistic circuits such as induction heads that enable pattern completion (Olsson et al., 2022). Our focus is complementary: rather than explaining *why* ICL works, we build practical predictors for *when* a particular prompt context will work.

### 2.2. Demonstration Selection and Prompt Optimization

Most demonstration selection methods aim to *find* a good context $D^\star$ for a query $q$. Early approaches select demonstrations via heuristic similarity (e.g., BM25 or embedding $k$NN), and study what makes an example effective for ICL (Liu et al., 2022). Learned retrievers and rankers train instance-adaptive selection policies from labeled data (Rubin et al., 2022; Ji et al., 2025). A complementary line performs model-based scoring of candidate demonstrations, using proxy criteria such as cross-entropy difference, influence estimates, or active selection to identify informative contexts (Iter et al., 2023; Nguyen & Wong, 2023; Zhang et al., 2022). More recently, LLM feedback and preferences have been used to guide demonstration selection (Zhang et al., 2025). Prompt engineering has also evolved toward programmatic optimization, where demonstrations and instructions are treated as tunable components in larger LLM programs (Zhou et al., 2022; Khattab et al., 2023; Opsahl-Ong et al., 2024). Orthogonal to *which* demonstrations are chosen, order sensitivity is a persistent challenge; methods such as Batch-ICL reduce order effects by decoupling demonstrations into independent forward passes (Zhang et al., 2024).

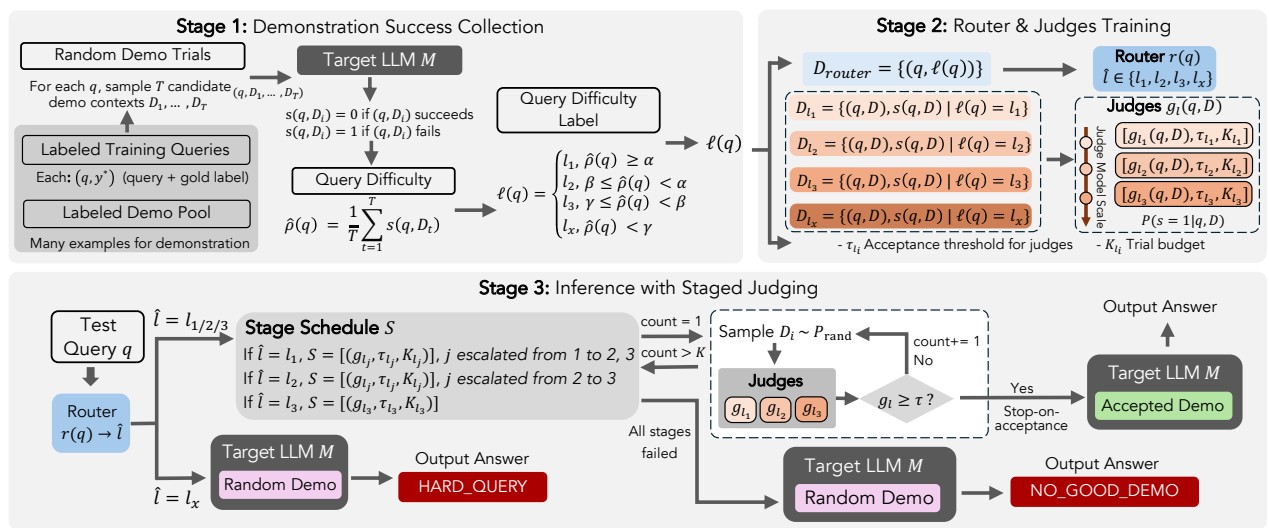

*Figure 2.* Overview of DiSP. **Stage 1:** run the target LLM on each training query under multiple random $k$-shot contexts to label success and estimate an empirical success rate for difficulty stratification. **Stage 2:** train a router and level-specific judges to predict success for a given $(q, D)$ pair. **Stage 3:** at test time, route each query and apply stop-on-acceptance sample-and-judge over sampled contexts up to a budget, emitting risk tags and falling back to a random context when needed.

## 2.3. Success Prediction, Routing, and Reliability

Most demonstration selection methods above aim to *find* a good context $D^\star$ for each query. In contrast, our focus is on *judging*: given a specific pair $(q, D)$, we predict whether ICL will succeed, turning selection into feasibility testing over proposed candidates. More broadly, an emerging perspective is to *judge* whether a model will succeed for a *given* input–context pair (e.g., $(q, D)$ in ICL), and to use such predictions to screen or prioritize candidate prompts under strict inference budgets. This is closely related to budgeted inference via routing/model cascades, where additional computation is allocated only when needed. Reliability is also a central concern: conformal prediction has been explored for uncertainty quantification in LLM settings (Xu & Lu, 2025; Vishwakarma et al., 2025), providing principled abstention mechanisms complementary to context selection.

## 3. Difficulty-Stratified Success Prediction for In-Context Learning

In this section, we propose DiSP (**Di**fficulty-Stratified **S**uccess **P**rediction), a framework for ICL under an inference budget. Rather than learning a task-specific retriever/ranker to find demonstrations, we learn efficient difficulty-stratified success predictors that *judge* whether a sampled context will work, as illustrated in Figure 2.

### 3.1. Problem Setup

Let $M$ be a frozen LLM and let $\mathcal{Y}$ be a finite label set rendered as short answer strings. For a query $q$ with gold

label $y^\star \in \mathcal{Y}$ and a $k$-shot demonstration *sequence* $D = (d_1, \ldots, d_k)$ sampled from a labeled pool $\mathcal{P}$, where each $d_i = (x_i, y_i)$, we form a prompt $\pi(q, D)$ by concatenating demonstrations in the given order followed by the query. The model predicts

$$\hat{y}(q, D) \;=\; \arg\max_{y \in \mathcal{Y}} \log p_M(y \mid \pi(q, D)), \qquad (2)$$

and we define the ICL success indicator

$$s(q, D) \;=\; \mathbb{I}[\hat{y}(q, D) = y^\star]. \qquad (3)$$

### 3.2. Random Demo Trials and Query Difficulty

Our method does not design a demonstration generator or retriever. Instead, we assume a random distribution over class-balanced demonstration contexts. For classification, we set $k = |\mathcal{Y}|$ and sample one labeled example per class uniformly from the pool, then randomize the order. This induces a proposal distribution $D \sim \mathcal{P}_{\text{rand}}$.

Using the LLM as an oracle during demonstration data construction, we estimate the empirical ICL success rate under random contexts:

$$\hat{\rho}(q) \;=\; \frac{1}{T} \sum_{t=1}^{T} s(q, D_t). \qquad (4)$$

where $T$ is the number of random demo trials used to compute $\hat{\rho}(q)$, a finite-sample approximation to the (unknown) true success rate $\rho(q)$ of query $q$ under $\mathcal{P}_{\text{rand}}$. We then assign each query to one of four difficulty levels

$\ell(q) \in \{l_1, l_2, l_3, l_x\}$:

$$\ell(q) = \begin{cases} l_1, & \hat{\rho}(q) \geq \alpha, \\ l_2, & \beta \leq \hat{\rho}(q) < \alpha, \\ l_3, & \gamma \leq \hat{\rho}(q) < \beta, \\ l_x, & \hat{\rho}(q) < \gamma, \end{cases} \quad (5)$$

where $(\alpha, \beta, \gamma)$ are thresholds with $\alpha > \beta > \gamma$.

Intuitively, $l_1$ queries are easy where random trials succeed frequently, $l_2$ queries benefit from a small amount of context trials, $l_3$ queries are brittle and require larger sampling budgets, and $l_x$ queries are unlikely to be solved by random demonstrations within a practical budget.

**Why four levels and level-specific judges?** The strata are a deployment-motivated discretization of the random-context success rate under the fixed proposal distribution. They correspond to distinct compute regimes: for $l_1$, a workable context is common and there is little value in spending large budgets; for $l_2/l_3$, workable contexts exist but are rarer and selection under a limited budget matters; for $l_x$, the success rate is so small that additional sampling and judging is typically inefficient. We therefore train judges only for $l_1/l_2/l_3$ and treat $l_x$ as an explicit risk stratum handled by a single-call fallback with a HARD_QUERY tag. Across $l_1/l_2/l_3$, we use increasing judge capacity so that expensive judging is invoked only for brittle queries, reducing the average inference overhead compared to a single worst-case judge used for every query.

**A coverage view for selecting $K$ for inference.** With the random-context success rate $\hat{\rho}(q)$, if we sample $K$ independent demo contexts $D_1, \ldots, D_K \sim \mathcal{P}_{\text{rand}}$, then the demo set contains at least one successful context with probability $1 - (1 - \hat{\rho}(q))^K$. This basic fact motivates level-dependent budgets $K_{l_1} < K_{l_2} < K_{l_3}$: for queries with larger $\hat{\rho}(q)$, a small $K$ already makes a good context likely to exist, whereas brittle queries require larger $K$ to reliably include a workable context. We develop this view and its implications for budgeting in Appendix A.1 and empirically study the $K$–accuracy trade-off in Section 4.7. Although $\hat{\rho}(q)$ is estimated from only $T$ Bernoulli trials, standard concentration bounds imply that level assignments are stable away from the thresholds $(\alpha, \beta, \gamma)$; see Appendix A.2.

**Judging as feasibility testing.** At inference, the judge is used as a *feasibility test* rather than a ranker. For each sampled demo context $D_i$, we compute a judge score $g_\ell(q, D_i)$ and compare it to a threshold $\tau_\ell$: if $g_\ell(q, D_i) \geq \tau_\ell$, we accept the demo and stop (stop-on-acceptance); otherwise we reject it and keep sampling, up to the budget $K_\ell$. If no candidate passes the test within the allotted trials, we emit the NO_GOOD_DEMO tag and query the LLM with a randomly sampled demo.

Judging is feasible because successful and failed demo contexts $(q, D)$ exhibit separable structure in representation space. In Section 4.3, we provide empirical evidence by training lightweight hidden-state probes: a simple MLP applied to the LLM hidden representations can distinguish successful from failed ICL trials with strong AUROC/AUPRC. This motivates learning compact proxy judges that predict feasibility from $(q, D)$ without the target LLMs. Under a bounded score error assumption, stop-on-acceptance guarantees that accepted contexts have true success probability at least $\tau_\ell$ up to the error tolerance, and NO_GOOD_DEMO is sound on the sampled candidate set (Lemmas A.4 and A.5). Formal statements are given in Appendix A.3.

### 3.3. Training Data Construction

For each training query $q$, we collect the tuple $\left(q, \{(D_t, s(q, D_t))\}_{t=1}^T\right)$ and compute $\ell(q)$ by (5). Concretely, for each $q$ we sample $T$ random contexts $D_1, \ldots, D_T \sim \mathcal{P}_{\text{rand}}$, run the LLM to obtain $\{s(q, D_t)\}_{t=1}^T$, compute $\hat{\rho}(q)$ by (4), and assign $\ell(q) \in \{l_1, l_2, l_3, l_x\}$ by thresholds.

This yields four supervised datasets:

$$\mathcal{D}_{\text{route}} = \{(q, \ell(q))\}, \quad (6)$$
$$\mathcal{D}_{l_1} = \{((q, D), s(q, D)) : \ell(q) = l_1\}, \quad (7)$$
$$\mathcal{D}_{l_2} = \{((q, D), s(q, D)) : \ell(q) = l_2\}, \quad (8)$$
$$\mathcal{D}_{l_3} = \{((q, D), s(q, D)) : \ell(q) = l_3\}. \quad (9)$$

We train a *router* on $\mathcal{D}_{\text{route}}$ and train level-specific *judges* on $\mathcal{D}_{l_1}$, $\mathcal{D}_{l_2}$ and $\mathcal{D}_{l_3}$.

### 3.4. Router and Level-Specific Judges

**Router $r(q)$.** The router predicts $\hat{\ell} = r(q)$ from the query text alone via four-way classification ($l_1/l_2/l_3/l_x$). This implements the principle "look at $q$ first": $\hat{\ell}$ determines whether to apply judging and, when applicable, which level-specific judge $g_{\hat{\ell}}$ to use.

**Judges $g_l(q, D)$.** The judges predict a success probability for a specific pair $(q, D)$:

$$g_\ell(q, D) \approx P(s(q, D) = 1 \mid q, D), \quad \ell \in \{l_1, l_2, l_3\}. \quad (10)$$

Both router and judges are encoders with classification heads. We use a base-size encoder for $g_{l_1}$, a larger one for $g_{l_2}$, and for $g_{l_3}$ we attach a lightweight head on top of representations from the target LLMs, matching the increased brittleness of $l_3$ contexts (details in Section 4). This keeps judging cheap on the many easy queries while reserving the most expensive judge features for the brittle stratum where they matter most.

---

**Algorithm 1** DiSP with difficulty-stratified judging

---

**Require:** Query $q$; LLM $M$; demo pool $\mathcal{P}$; budgets $K_{l_1}, K_{l_2}, K_{l_3}$; thresholds $\tau_{l_1}, \tau_{l_2}, \tau_{l_3}$

1:  $\hat{\ell} \leftarrow r(q)$
2:  **if** $\hat{\ell} = l_x$ **then**
3:     Sample $D \sim \mathcal{P}_{\text{rand}}$ from $\mathcal{P}$
4:     **return** $M(\pi(q, D))$ with tag HARD_QUERY
5:  **else**
6:     **for** $i = 1$ to $K_{\hat{\ell}}$ **do**
7:        Sample $D_i \sim \mathcal{P}_{\text{rand}}$ from $\mathcal{P}$
8:        $p_i \leftarrow g_{\hat{\ell}}(q, D_i)$
9:        **if** $p_i \geq \tau_{\hat{\ell}}$ **then**
10:          **return** $M(\pi(q, D_i))$ {stop-on-accept}
11:        **end if**
12:     **end for**
13:     Sample $D \sim \mathcal{P}_{\text{rand}}$ from $\mathcal{P}$
14:     **return** $M(\pi(q, D))$ with tag NO_GOOD_DEMO
15: **end if**

---

### 3.5. Test-Time Workflow for Each Query

As illustrated in Figure 2, we first route the query with $\hat{\ell} \leftarrow r(q)$. If $\hat{\ell} = l_x$, we query the LLM once with a randomly sampled context and emit HARD_QUERY. Otherwise ($\hat{\ell} \in \{l_1, l_2, l_3\}$), we run stop-on-acceptance with judge $g_{\hat{\ell}}$, threshold $\tau_{\hat{\ell}}$, and budget $K_{\hat{\ell}}$; if no candidate passes within budget, it turns to a random context and emit NO_GOOD_DEMO (see Appendices A.1 and A.3 for a motivating analysis). Algorithm 1 summarizes the procedure. At inference, additional computation comes from the lightweight router/judges. The sampling budgets $K_\ell$ provide a direct accuracy–cost knob, and the level-dependent design allocates more compute only to queries in need.

## 4. Experiments

### 4.1. Experimental Setup

**Datasets and Backbones.** We evaluate on five text classification benchmarks spanning question classification, sentiment analysis, topic classification, and natural language inference: (1) TREC (Li & Roth, 2002) (6-way question type classification), (2) SST-2 and (3) SST-5 from the Stanford Sentiment Treebank (Socher et al., 2013), (4) AG-NEWS (Zhang et al., 2015) (4-way news topic), and (5) MNLI (Williams et al., 2018) (3-way natural language inference). We experiment with two open-source backbones: Llama 3–8B (Meta AI, 2024) and Qwen 2.5–7B (Qwen Team, 2024). We use $k$-shot ICL with $k$ set to the number of classes (one demonstration per label) for all classification tasks. Prompt templates and label verbalizers are dataset-specific and provided in Appendix D.

**Baselines.** We compare against (1) zero-shot (no demonstrations) and the few-shot baselines, including (2) a single random demonstration context (Random), (3) a retrieval-based context (BM25), and recent demonstration selection baselines (4) Uprise (Cheng et al., 2023), (5) $\text{Se}^2$ (Liu et al., 2024), and (6) SeDPO (Ji et al., 2025). For fair comparisons, each method selects a single context per query.

**Router and judges.** As in Section 3.2, DiSP trains router and judges to predict the difficulty level of $q$ and success outcome of $(q, D)$ pairs. We implement the router $r(q)$ with a BERT-base ($\sim$110M parameters). For judges, we use (i) a BERT-base ($\sim$110M) for $g_{l_1}$, (ii) a RoBERTa-Large ($\sim$330M) for $g_{l_2}$, and (iii) a lightweight classification head over representations from the target LLM for $g_{l_3}$. In all cases, judges use binary classification heads and the target LLM is kept frozen. Details of training and all hyperparameters are provided in Appendix B.

**Evaluation metrics.** We report accuracy on each test set compared with baselines. To quantify efficiency, we report *GPU time* using NVIDIA A100 GPUs for all methods. We report (i) training cost, including supervision collection and parameter training, and (ii) test cost, including demonstration selection and the final target-LLM inference.

### 4.2. Main Results

**Accuracy.** Table 1 reports accuracy on five datasets with two backbone LLMs. Across both backbones, DiSP achieves the best average accuracy, for example, 77.6% (+11.2 over zero-shot) on LLaMA3-8B and 82.9% (+13.1) on Qwen2.5-7B. The gains are largest on harder datasets where demonstrations and selection matter most (e.g., TREC and MNLI), while improvements are modest on already-easy settings such as SST-2 (zero-shot is $\geq$94.7%). Compared with strong selection baselines (BM25/Uprise/Se$^2$/SeDPO), DiSP improves by 2.8% over the strongest baseline on LLaMA3-8B and by 2.4% points on Qwen2.5-7B, while remaining competitive on the few settings where a baseline is slightly better (e.g., SST-5 on Qwen2.5-7B).

Notably, DiSP achieves these gains while using a fixed random proposal distribution and a single inference pipeline across datasets, without training task-specific retrievers or rankers. This suggests that judging as a feasibility test can capture a large fraction of the benefit of learned selection with substantially less specialization. Since DiSP involves random sampling of demonstrations, results in Table 1 are reported as the mean over 6 independent runs.

**Efficiency and wall-clock cost.** Beyond accuracy, we compare end-to-end efficiency against baselines. Table 2 summarizes wall-clock cost averaged over five datasets.

*Table 1.* Performance (%) on selected classification benchmarks with various demonstration selection methods. **Bold** indicates the best result.

| Model / Method | | Datasets | | | | | |
|---|---|---|---|---|---|---|---|
| | | **TREC** | **SST-2** | **SST-5** | **AGNews** | **MNLI** | **Avg** |
| **LLaMA3-8B** | | | | | | | |
| Zero-shot | | 70.5 | 94.7 | 41.7 | 72.9 | 52.3 | 66.4 |
| Few-shot | Random | 73.8 (+3.3) | 94.9 (+0.2) | 46.6 (+4.9) | 84.1 (+11.2) | 66.9 (+14.6) | 73.3 (+6.9) |
| | BM25 | 75.2 (+4.7) | 94.7 (+0.0) | 52.3 (+10.6) | 86.1 (+13.2) | 65.6 (+13.3) | 74.8 (+8.4) |
| | Uprise (Cheng et al. 2023) | 75.8 (+5.3) | 94.0 (-0.7) | 48.3 (+6.6) | **90.7** (+17.8) | 58.9 (+6.6) | 73.6 (+7.2) |
| | Se$^2$ (Liu et al. 2024) | 68.2 (-2.3) | 93.4 (-1.3) | 51.7 (+10.0) | 86.1 (+13.2) | 60.9 (+8.6) | 72.1 (+5.6) |
| | SeDPO (Ji et al. 2025) | 67.5 (-3.0) | 93.4 (-1.3) | 47.0 (+5.3) | 88.1 (+15.2) | 61.6 (+9.3) | 71.5 (+5.1) |
| | **DiSP** (Ours) | **79.2** (+8.7) | **95.4** (+0.7) | **54.3** (+12.6) | 89.7 (+16.8) | **69.5** (+17.2) | **77.6** (+11.2) |
| **Qwen2.5-7B** | | | | | | | |
| Zero-shot | | 63.1 | 95.4 | 42.4 | 71.5 | 76.8 | 69.8 |
| Few-shot | Random | 80.3 (+17.2) | 93.4 (-2.0) | 49.7 (+7.3) | 80.4 (+8.9) | 85.0 (+8.2) | 77.8 (+8.0) |
| | BM25 | 83.9 (+20.8) | 92.7 (-2.7) | **55.6** (+13.2) | 84.1 (+12.6) | 86.1 (+9.3) | 80.5 (+10.7) |
| | Uprise (Cheng et al. 2023) | 83.2 (+20.1) | 94.7 (-0.7) | 54.3 (+11.9) | **89.4** (+17.9) | 80.1 (+3.3) | 80.3 (+10.5) |
| | Se$^2$ (Liu et al. 2024) | 82.1 (+19.0) | 94.7 (-0.7) | 53.6 (+11.2) | 88.1 (+16.6) | 82.1 (+5.3) | 80.1 (+10.3) |
| | SeDPO (Ji et al. 2025) | 82.1 (+19.0) | **96.0** (+0.6) | 45.7 (+3.3) | 87.4 (+15.9) | 80.8 (+4.0) | 78.4 (+8.6) |
| | **DiSP** (Ours) | **87.3** (+24.2) | **96.0** (+0.6) | 54.3 (+11.9) | **89.4** (+17.9) | **87.4** (+10.6) | **82.9** (+13.1) |

*Table 2.* Wall-clock cost averaged over five datasets.

| Method | Time (min) | Relative (vs. DiSP) |
|---|---|---|
| *Training cost* | | |
| Uprise | 13.4 | 2.0× |
| Se$^2$ | 114.9 | 17.1× |
| SeDPO | 157.1 | 23.4× |
| DiSP | 6.7 | 1.0× |
| *Test cost* | | |
| Uprise | 0.4 | 4.0× |
| Se$^2$ | 0.6 | 6.0× |
| SeDPO | 0.6 | 6.0× |
| DiSP | 0.1 | 1.0× |
| *Total cost* | | |
| Uprise | 13.8 | 2.0× |
| Se$^2$ | 115.4 | 17.0× |
| SeDPO | 157.6 | 23.2× |
| DiSP | 6.8 | 1.0× |

DiSP is consistently lightweight: its training pipeline takes 6.7 minutes on average, versus 13.4 minutes for Uprise (2.0×), 114.9 minutes for Se$^2$ (17.1×), and 157.1 minutes for SeDPO (23.4×). At test time, DiSP adds minimal overhead (0.1 minutes), while methods that rely on retrieval and/or heavier offline artifacts are 4–6× slower on average. Overall, DiSP reduces total cost to 6.8 minutes, compared to 13.8 (Uprise), 115.4 (Se$^2$), and 157.6 (SeDPO), highlighting that the sample-and-test pipeline can improve accuracy while remaining inexpensive to run. The complete per-dataset timing breakdown is provided in Appendix B.2.

### 4.3. Separability Evidence from Representation Probes

This study is motivated by a simple question: *can a classifier distinguish successful from failed ICL trials from the specific representation of a $(q, D)$ pair?* Concretely, we extract last-layer hidden-state vectors for prompt instances $x = (q, D)$ from target LLMs, label them with the success indicator $s(q, D)$, and train an MLP probe to predict success.

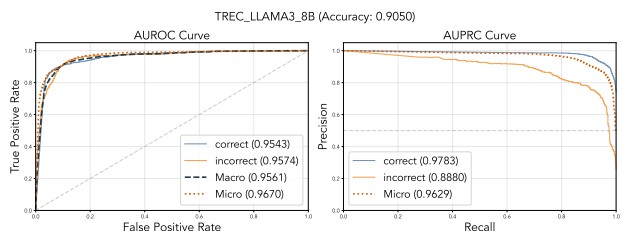

*Figure 3.* Hidden-state probes provide evidence that success and failure form separable clusters in the representation space (LLaMA3-8B on TREC).

We report AUROC/AUPRC in Figure 3 for LLaMA3-8B on TREC. Complete results for all datasets/backbones are provided in Appendix C. Overall, the probe achieves strong separability across tasks, supporting the feasibility of success prediction with a cheaper framework.

### 4.4. Difficulty Composition and Success Rates

We report the fraction of test queries assigned to each difficulty level ($l_1/l_2/l_3/l_x$), using *oracle strata* computed from $\hat{\rho}(q)$ on held-out data with the target LLMs. Table 3 shows that difficulty composition varies substantially across datasets and backbones. For example, SST-2 is dominated

*Table 3.* Oracle difficulty composition (%) and random-context success rates (%) on test sets. For each query, we sample 1,000 random demonstrations and report mean accuracy within each oracle stratum.

| | Dataset | Composition (%) | | | | Success rate (%) | | | |
|---|---|---|---|---|---|---|---|---|---|
| | | $l_1$ | $l_2$ | $l_3$ | $l_x$ | $l_1$ | $l_2$ | $l_3$ | $l_x$ |
| **LLaMA** | TREC | 64.4 | 22.1 | 6.0 | 7.4 | 96.2 | 48.8 | 21.1 | 3.2 |
| | SST-2 | 94.0 | 0.7 | 0.0 | 5.3 | 99.5 | 50.0 | 19.1 | 7.9 |
| | SST-5 | 43.7 | 9.3 | 8.6 | 38.4 | 96.1 | 48.2 | 20.8 | 1.4 |
| | AGNEWS | 81.5 | 4.6 | 2.0 | 11.9 | 98.8 | 44.3 | 18.3 | 1.7 |
| | MNLI | 62.9 | 6.6 | 4.6 | 25.8 | 98.1 | 41.0 | 22.1 | 1.5 |
| **Qwen** | TREC | 73.2 | 14.1 | 4.0 | 8.7 | 97.3 | 56.2 | 20.0 | 2.3 |
| | SST-2 | 92.1 | 2.0 | 1.9 | 4.0 | 99.5 | 50.0 | 21.7 | 1.7 |
| | SST-5 | 43.7 | 11.2 | 4.0 | 41.1 | 98.5 | 55.6 | 22.5 | 1.2 |
| | AGNEWS | 78.1 | 6.0 | 2.0 | 13.9 | 99.0 | 48.9 | 17.3 | 0.2 |
| | MNLI | 84.1 | 3.4 | 2.6 | 9.9 | 98.8 | 55.0 | 19.9 | 2.0 |

*Table 4.* Per-stratum accuracy (%) grouped by router-predicted difficulty, with zero-shot analysis for $l_1$ queries. $R_{\text{zero/ICL}}$ denotes the ratio of zero-shot to DiSP accuracy on $l_1$.

| | Dataset | DiSP accuracy by stratum | | | | 0-shot on $l_1$ | |
|---|---|---|---|---|---|---|---|
| | | $l_1$ | $l_2$ | $l_3$ | $l_x$ | Acc | $R$ |
| **LLaMA** | TREC | 90.4 | 61.5 | 0.0 | 0.0 | 76.9 | 0.851 |
| | SST-2 | 95.7 | 100.0 | 0.0 | 0.0 | 95.0 | 0.993 |
| | SST-5 | 73.9 | 64.7 | 47.3 | 0.0 | 56.5 | 0.765 |
| | AGNEWS | 94.3 | 71.9 | 100.0 | 50.0 | 88.7 | 0.940 |
| | MNLI | 82.1 | 60.3 | 70.4 | 0.0 | 56.4 | 0.688 |
| **Qwen** | TREC | 92.7 | 72.7 | 100.0 | 22.2 | 67.5 | 0.728 |
| | SST-2 | 96.5 | 100.0 | 66.7 | 0.0 | 95.1 | 0.985 |
| | SST-5 | 74.1 | 50.0 | 50.5 | 33.3 | 51.9 | 0.700 |
| | AGNEWS | 94.0 | 71.4 | 71.4 | 14.3 | 88.8 | 0.945 |
| | MNLI | 91.5 | 75.0 | 85.2 | 0.0 | 80.2 | 0.876 |

by $l_1$ queries (92–94%) with only a small $l_x$ tail (4–5%), whereas SST-5 has only 43.7% $l_1$ and a large $l_x$ fraction (38–41%). MNLI also exhibits a strong backbone effect, with $l_x$ dropping from 25.8% (LLaMA3-8B) to 9.9% (Qwen2.5-7B). This backbone dependence cautions against treating demonstration selection primarily as a cross-model generalization mechanism: the difficulty profile is not invariant across target LLMs.

The table also shows the mean success rate under 1,000 random demonstrations for each stratum, confirming that difficulty correlates strongly with success: $l_1$ strata achieve near-ceiling performance (96–99%), while $l_x$ drops to single digits. This validates the operational meaning of our stratification and motivates allocating larger sampling budgets only to queries most likely to benefit.

### 4.5. Auxiliary Predictors: Router and Judges

We analyze the performance of the router and judges: the router results serve as an indirect indicator, while the precision of judges in identifying correct demonstrations largely determines the final performance.

**Router evaluation.** Router accuracy varies substantially across scenarios, ranging from 48.5% (SST-5, Qwen) to 88.7% (SST-2, LLaMA), reflecting that adjacent-stratum boundaries are more ambiguous under some scenarios than others. Full per-scenario router accuracy results and analysis are reported in Appendix B.1.

**Judge evaluation.** Adopting accuracy as a metric to evaluate judges can be misleading, since the objective of DiSP is not to perfectly classify success/failure for arbitrary $(q, D)$ pairs, but to efficiently discriminate demonstrations that make the target LLM answer correctly via sample-and-test. In this setting, what matters most is the *precision* of accepted demonstrations (i.e., how often a judge-accepted context actually yields a correct target response), while recall is secondary as long as it is not so low that the sampler

frequently exhausts its budget. Accordingly, we focus on judge performance within each difficulty stratum, which is discussed in Section 4.6.

### 4.6. Per-stratum Accuracy and Zero-shot Analysis

We report per-stratum accuracy of DiSP by the *router-predicted* difficulty level $\hat{\ell} = r(q)$. This breakdown reflects how accuracy varies from easy ($l_1$) to brittle ($l_2/l_3$) and very hard ($l_x$) queries. As shown in Table 4, accuracy is consistently high on predicted $l_1$ queries. Predicted $l_2$ strata exhibit lower but still substantial accuracy, matching the intended regime where judging and limited sampling are most beneficial. In contrast, performance drops sharply for predicted $l_3$ and especially $l_x$, denoting queries for which workable random demonstrations are rare under the proposal and additional sampling is less likely to pay off. We note that some extreme values (0 or 100%) can occur when a predicted stratum has very small support on a given scenario. Importantly, accuracies on the harder predicted strata ($l_2/l_3/l_x$) are substantially higher than the corresponding oracle random-context success rates in Table 3, providing further evidence that DiSP can identify helpful contexts beyond what is achievable by random demonstrations alone.

Furthermore, we analyze a zero-shot variant on $l_1$ queries. Table 4 reports the ratio $R_{\text{zero/ICL}} = \text{Acc}_{\text{zero}}/\text{Acc}_{\text{ICL}}$ to quantify the impact. Zero-shot recovers most of the ICL performance on routed $l_1$ queries for SST-2 ($R \approx 0.99$), indicating that demonstrations can be skipped with minimal loss in this setting. However, on TREC and SST-5 the gap is larger ($R \approx 0.70$–$0.85$), suggesting that our $l_1$ stratum captures queries that are *easy under random ICL contexts* rather than universally easy in the zero-shot regime.

### 4.7. Budgeted Sampling with Random Demonstrations

We characterize how much sampling DiSP actually performs under the deployed, stratum-dependent budgets $K_{\hat{\ell}}$ (with $\hat{\ell} = r(q)$) using two complementary statistics in Table 5: (i)

rejection rate, the fraction of queries that exhaust the full budget $K_{\hat{\ell}}$ without any candidate being accepted (triggering the fallback), and (ii) average attempts, the mean number of candidates tested before acceptance, computed only over non-rejected queries. Overall, acceptance typically happens within a small number of trials for predicted $l_1/l_2$ (often $\approx$1–2 attempts), indicating that the sample-and-test procedure rarely needs to approach the budget on easier routed queries. In contrast, predicted $l_3$ exhibits higher rejection and/or more attempts (e.g., LLaMA on SST-5: 14.3% rejection with 4.8 attempts among accepted; Qwen on AGNEWS: 28.6% rejection), reflecting that workable random demonstrations are rarer in this regime. N/A indicates that a stratum has no support for a given dataset/backbone setting (i.e., no queries are routed to that bucket), so the corresponding statistic is undefined.

*Table 5.* Rejection rate (%) and average attempts by predicted difficulty. Rej. rate: fraction of queries exhausting budget $K_{\hat{\ell}}$ without acceptance; Avg. att.: mean attempts before acceptance (non-rejected only).

| | Dataset | Rej. rate (%) | | | Avg. att. | | |
|---|---|---|---|---|---|---|---|
| | | $l_1$ | $l_2$ | $l_3$ | $l_1$ | $l_2$ | $l_3$ |
| **LLaMA** | TREC | N/A | N/A | N/A | N/A | N/A | N/A |
| | SST-2 | N/A | N/A | N/A | 1.2 | 1.0 | N/A |
| | SST-5 | 8.7 | N/A | 14.3 | 1.9 | 4.8 | 4.8 |
| | AGNEWS | N/A | N/A | N/A | 1.0 | 1.9 | 3.0 |
| | MNLI | 5.1 | 1.7 | 1.9 | 1.0 | 1.7 | 2.3 |
| **Qwen** | TREC | N/A | N/A | N/A | 1.1 | 2.6 | 9.5 |
| | SST-2 | N/A | N/A | N/A | 1.0 | 1.0 | 2.0 |
| | SST-5 | N/A | 3.3 | 1.1 | 1.0 | 2.0 | 3.9 |
| | AGNEWS | N/A | N/A | 28.6 | 1.0 | 6.3 | 3.0 |
| | MNLI | N/A | N/A | N/A | 1.3 | 2.5 | 2.6 |

## 4.8. Risk Tagging: HARD_QUERY and NO_GOOD_DEMO

In Section 3.5, we have defined two diagnostic signals and treat them jointly as a single *risk-tag* for analysis. HARD_QUERY flags queries that the router predicts as $l_x$, indicating that additional random sampling is unlikely to help within the configured budget. NO_GOOD_DEMO flags queries for which none of the sampled contexts is predicted to succeed within the judge budgets. Across all scenarios, only 2.2% of queries receive these tags. 88.3% of risk-tagged queries are oracle-labeled as $l_x$. This suggests that our risk-tagging is a high-precision indicator of the extreme tail of difficulty, and points to a simple mechanism for handling high-risk queries in practice (e.g., abstaining/refusing, requesting clarification, deferring to a stronger model, or escalating to human review) while avoiding wasted sampling on cases where random ICL is unlikely to succeed. We emphasize that this diagnostic is intended to be conservative: high precision does not guarantee high recall, and some oracle-$l_x$ queries may remain untagged.

## 4.9. Cross-Dataset Training with a Unified Judge

The judges in DiSP solve a binary prediction problem—whether a sampled query–context pair $(q, D)$ will succeed—and are not tied to a particular label set. We therefore study whether they can be trained jointly across tasks. For a fixed target backbone, we construct a pooled training set by taking the union of all five datasets.

Table 6 shows that pooling is largely harmless on easy queries ($l_1$) for LLaMA: the overall drop is $< 1\%$ (91.5$\rightarrow$90.6), suggesting strong transfer in the regime where success is less sensitive to fine-grained dataset structure. On $l_2$, pooling is mixed: LLaMA drops by 5.2% overall, with MNLI slightly improving. The largest degradation appears on $l_3$, driven by SST-5 and MNLI. These patterns indicate that the hardest stratum is more dataset-specific and benefits from in-domain supervision, while unified training remains a promising direction for reducing the number of judges when scaling DiSP to many tasks. We note that some $l_3$ subsets are small, so dataset-level differences should be interpreted with caution.

*Table 6.* Cross-dataset training of judges. Entries report accuracy (%) for **Sep.** (per-dataset judges) and **Pool.** (unified judges trained on the union of all datasets). **All** aggregates test queries pooled across all five datasets within the stratum.

| Stratum | Dataset | LLaMA3-8B | | Qwen2.5-7B | |
|---|---|---|---|---|---|
| | | Sep. | Pool. | Sep. | Pool. |
| $l_1$ | TREC | 90.4 | 90.4 | 92.7 | 91.9 |
| | SST-2 | 95.7 | 95.0 | 96.5 | 96.5 |
| | SST-5 | 73.9 | 65.2 | 74.1 | 74.1 |
| | AGNEWS | 94.3 | 93.4 | 94.0 | 94.0 |
| | MNLI | 82.1 | 82.1 | 91.5 | 90.6 |
| | **All** | 91.5 | 90.6 | 92.8 | 92.4 |
| $l_2$ | TREC | 61.5 | 58.7 | 72.7 | 72.7 |
| | SST-2 | 100.0 | 100.0 | 100.0 | 100.0 |
| | SST-5 | 64.7 | 58.8 | 50.0 | 56.7 |
| | AGNEWS | 71.9 | 65.6 | 71.4 | 57.1 |
| | MNLI | 60.3 | 62.1 | 75.0 | 68.8 |
| | **All** | 65.7 | 60.5 | 65.7 | 65.7 |
| $l_3$ | TREC | 0.0 | 0.0 | 100.0 | 100.0 |
| | SST-2 | – | – | 66.7 | 66.7 |
| | SST-5 | 47.3 | 47.3 | 50.6 | 40.7 |
| | AGNEWS | 100.0 | 100.0 | 71.4 | 71.4 |
| | MNLI | 70.4 | 61.1 | 85.2 | 74.1 |
| | **All** | 55.8 | 52.4 | 61.7 | 53.2 |

## 5. Conclusion

We introduced DiSP, a judge-centric framework for demonstration selection in ICL under an explicit inference budget. DiSP treats selection as feasibility testing: it stratifies

queries by difficulty via offline random-demo trials, routes each query to a compute regime, and applies level-specific judges with stop-on-acceptance over sampled contexts (with a risk-tagged fallback when no good context is found). Across five classification benchmarks with Llama 3–8B and Qwen 2.5–7B, DiSP achieves the best average accuracy while reducing end-to-end wall-clock cost by up to $23\times$. Future work includes improving calibration/stopping, reducing offline labeling cost, and extending beyond classification.

## Acknowledgements

This work was supported in part by the National Natural Science Foundation of China [62576126]; and the Key R&D Program of Heilongjiang Province [2023ZX01A11].

## Limitations

**Offline labeling cost.** Our supervision is obtained by running the target LLM (as a labeling oracle) across multiple random demonstration contexts per training query. While amortized across deployments, this upfront cost can be non-trivial when scaling to larger datasets, many prompt variants, or many backbones. This challenge is not unique to DiSP: data-driven demonstration selection methods generally rely on offline data collection and training.

**Dependence on the proposal distribution.** Judges are trained on $(q, D)$ pairs induced by a specific proposal distribution (random composition in this study). Changing the proposal, demo pool, or prompt format can shift the $(q, D)$ distribution, degrading reliability and typically requiring re-collection.

## Impact Statement

This paper presents DiSP, which routes queries by difficulty and uses lightweight judges to predict whether a sampled demonstration context is likely to succeed for a fixed target LLM. By reducing the need to evaluate many candidate contexts with the target LLM, DiSP can lower test-time latency and computational costs.

However, making ICL more efficient may also facilitate deployment at larger scale, including in harmful applications. Auxiliary predictors may inherit biases from the target LLM and data, and distribution shift (e.g., changing the demo pool, proposal distribution, or backbone) can lead to miscalibrated accept/reject decisions. Practitioners should curate demonstration pools and monitor for shifts in deployment; diagnostic tags (HARD_QUERY, NO_GOOD_DEMO) can support abstention or escalation policies but are not guarantees.

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

# A. Additional Theoretical Results

This appendix provides detailed derivations and proofs for the theoretical statements used to motivate our design choices: (i) how the random-context success rate $\rho(q)$ determines the sampling budget $K$, (ii) how finite random-demo trials affect the stability of difficulty labels, and (iii) how judge accuracy translates into guarantees for thresholding, stop-on-acceptance, and risk tags.

## A.1. Coverage of Random Candidates and Budget Selection

**Setup.** Fix a query $q$ and consider contexts drawn i.i.d. from a proposal distribution $D \sim \mathcal{P}_{\text{rand}}$. Recall the ICL success indicator $s(q, D) \in \{0, 1\}$. Define the random-context success rate

$$\rho(q) \;=\; \Pr_{D \sim \mathcal{P}_{\text{rand}}} [s(q, D) = 1]. \tag{11}$$

**Proposition A.1** (Candidate coverage under random sampling). *Let $D_1, \ldots, D_K \overset{\text{i.i.d.}}{\sim} \mathcal{P}_{\text{rand}}$. Then*

$$\Pr(\exists i \leq K : s(q, D_i) = 1) \;=\; 1 - (1 - \rho(q))^K. \tag{12}$$

*Proof.* Let $E_i$ be the event $\{s(q, D_i) = 0\}$. Since the $D_i$ are i.i.d., the events $E_i$ are independent and $\Pr(E_i) = 1 - \rho(q)$. Therefore,

$$\Pr(\forall i \leq K, \; s(q, D_i) = 0) \;=\; \prod_{i=1}^{K} \Pr(E_i) \;=\; (1 - \rho(q))^K.$$

Taking complements yields (12). $\square$

**Budgeting for a target miss probability.** If we want the miss probability to be at most $\delta \in (0, 1)$, i.e., $\Pr(\forall i \leq K, \; s(q, D_i) = 0) \leq \delta$, then by Proposition A.1 it suffices to choose

$$K \;\geq\; \frac{\log \delta}{\log(1 - \rho(q))}. \tag{13}$$

This expression is well-defined for $\rho(q) \in (0, 1)$ since $\log(1 - \rho(q)) < 0$.

**Stratum-wise worst-case budgeting.** Our difficulty thresholds in Equation (5) imply conservative lower bounds on $K$ for each stratum. If we treat $l_2$ queries as satisfying $\rho(q) \geq \beta$ and $l_3$ queries as satisfying $\rho(q) \geq \gamma$, then it suffices to choose

$$K_{l_2} \;\geq\; \frac{\log \delta}{\log(1 - \beta)}, \qquad K_{l_3} \;\geq\; \frac{\log \delta}{\log(1 - \gamma)}. \tag{14}$$

Importantly, these are *coverage* guarantees: they ensure that the sampled candidate set contains at least one successful context with high probability. Since our system selects contexts using a learned judge rather than scoring candidates with the task LLM, it must still *select* a good context from the sampled set; consequently, the end-to-end $K$ required in practice is typically larger. In our experiments, we use $K_{l_2} = 20$ and $K_{l_3} = 30$, which makes the worst-case miss probabilities $(1 - \beta)^{K_{l_2}}$ and $(1 - \gamma)^{K_{l_3}}$ under the stratum lower bounds.

## A.2. Concentration of $\hat{\rho}(q)$ and Stability of Difficulty Labels

**Setup.** For a fixed query $q$, let $D_1, \ldots, D_T \overset{\text{i.i.d.}}{\sim} \mathcal{P}_{\text{rand}}$ and define

$$\hat{\rho}(q) \;=\; \frac{1}{T} \sum_{t=1}^{T} s(q, D_t).$$

Then $\hat{\rho}(q)$ is the sample mean of $T$ i.i.d. Bernoulli random variables with mean $\rho(q)$.

**Lemma A.2** (Hoeffding bound for $\hat{\rho}(q)$). *For any $\varepsilon > 0$,*

$$\Pr(|\hat{\rho}(q) - \rho(q)| \geq \varepsilon) \;\leq\; 2 \exp(-2T\varepsilon^2). \tag{15}$$

*Proof.* This is a direct application of Hoeffding's inequality to bounded variables $s(q, D_t) \in [0, 1]$. $\square$

**A margin-based mislabel bound.** The difficulty thresholds $(\alpha, \beta, \gamma)$ in Equation (5) induce three boundary decisions: $\hat{\rho}(q) \geq \alpha$ vs. $< \alpha$, $\hat{\rho}(q) \geq \beta$ vs. $< \beta$, and $\hat{\rho}(q) \geq \gamma$ vs. $< \gamma$. The following corollary makes explicit that mislabeling probability decays exponentially in $T$ when $\rho(q)$ has margin from the thresholds.

**Corollary A.3** (Crossing a threshold is exponentially unlikely with margin). *Fix a threshold $t \in (0, 1)$.*

1. *If $\rho(q) \geq t + \Delta$ for some $\Delta > 0$, then $\Pr(\hat{\rho}(q) < t) \leq \exp(-2T\Delta^2)$.*

2. *If $\rho(q) \leq t - \Delta$ for some $\Delta > 0$, then $\Pr(\hat{\rho}(q) \geq t) \leq \exp(-2T\Delta^2)$.*

*Proof.* For (1), $\{\hat{\rho}(q) < t\} \subseteq \{\hat{\rho}(q) - \rho(q) \leq -\Delta\}$; apply Lemma A.2 to obtain $\Pr(\hat{\rho}(q) - \rho(q) \leq -\Delta) \leq \exp(-2T\Delta^2)$. □

The proof of (2) is analogous.

**Sample complexity for stable stratification.** As a simple implication, to ensure $\Pr(\hat{\rho}(q) < t) \leq \delta$ for all queries with $\rho(q) \geq t + \Delta$, it suffices to choose $T \geq \frac{1}{2\Delta^2} \log \frac{1}{\delta}$. If we want this to hold for *all* $n$ training queries simultaneously, a union bound yields $T \geq \frac{1}{2\Delta^2} \log \frac{n}{\delta}$. These bounds make the role of $T$ explicit: larger $T$ reduces label noise for all queries except those intrinsically close to the decision thresholds (e.g., $\rho(q) \approx \alpha$, $\rho(q) \approx \beta$, or $\rho(q) \approx \gamma$), for which any hard stratification is inherently ambiguous. In our experiments, we use $T = 20$ trials per training query as a compute–stability trade-off.

### A.3. Judging Guarantees for Thresholding and Stop-on-Acceptance

**Setup.** Fix a query $q$ and a candidate set $D_1, \ldots, D_K$. Let $p_i = p(q, D_i) = \Pr(s(q, D_i) = 1 \mid q, D_i)$ be the true success probability of each candidate and let $\hat{p}_i$ be the judge score (e.g., $\hat{p}_i = g_{\hat{\ell}}(q, D_i)$). Let $\tau \in (0, 1)$ be the acceptance threshold. Our inference rule scans candidates and *accepts the first* index $i$ for which $\hat{p}_i \geq \tau$ and stops (stop-on-acceptance); if no candidate is accepted within the budget, it emits NO_GOOD_DEMO and queries the LLM with a randomly sampled context. The lemmas below apply to this thresholded, stop-on-acceptance procedure.

**Feasibility testing vs. ranking.** This design treats judging as a feasibility test around a decision threshold rather than a full ranking problem. The threshold $\tau$ trades off two failure modes: setting it too low increases false positives (accepting contexts that still fail), while setting it too high increases false negatives (rejecting workable contexts and exhausting the budget).

**Soundness of NO_GOOD_DEMO.** Let $\tau \in (0, 1)$ be the threshold used by the inference rule in Algorithm 1. The next lemma formalizes the idea that thresholding is robust to small judge error.

**Lemma A.4** (NO_GOOD_DEMO is sound up to $\varepsilon$). *Assume $|\hat{p}_i - p_i| \leq \varepsilon$ for all $i$.*

1. *If $\max_i \hat{p}_i < \tau$, then $\max_i p_i < \tau + \varepsilon$.*

2. *If there exists $j$ with $p_j \geq \tau + \varepsilon$, then $\max_i \hat{p}_i \geq \tau$.*

*Proof.* For (1), for every $i$, $p_i \leq \hat{p}_i + \varepsilon < \tau + \varepsilon$, hence $\max_i p_i < \tau + \varepsilon$. For (2), pick $j$ with $p_j \geq \tau + \varepsilon$; then $\hat{p}_j \geq p_j - \varepsilon \geq \tau$, hence $\max_i \hat{p}_i \geq \hat{p}_j \geq \tau$. □

**Stop-on-acceptance guarantee.**

**Lemma A.5** (Accepted contexts have bounded true success probability). *Assume $|\hat{p}_i - p_i| \leq \varepsilon$ for all $i$ and suppose the inference rule returns some index $i$ with $\hat{p}_i \geq \tau$. Then $p_i \geq \tau - \varepsilon$.*

*Proof.* Immediate from $p_i \geq \hat{p}_i - \varepsilon \geq \tau - \varepsilon$. □

**Remarks.** The lemmas above are stated under a deterministic, uniform error condition for clarity and match the thresholded, stop-on-acceptance procedure used by our system. In practice, judge errors are random and depend on $(q, D)$; one can obtain high-probability versions by bounding the maximum error over the $K$ sampled candidates (e.g., via union bounds) or by working with scores and conformal-style guarantees. We leave such refinements to future work and focus on empirical evaluation.

*Table 7.* Key hyperparameters used in our experiments.

| Component | Setting |
|---|---|
| ICL shots | $k = |\mathcal{Y}|$ (one demo per class) |
| Random-demo trials ($\hat{\rho}$) | $T = 20$ per training query |
| Difficulty thresholds | $\alpha = 0.75, \beta = 0.3, \gamma = 0.1$ |
| Inference budgets | $K_{l_1} = 10, K_{l_2} = 20, K_{l_3} = 30$ |
| Acceptance thresholds | tune $\tau_{l_1}, \tau_{l_2}, \tau_{l_3}$ on validation sets under each scenario |

## A.4. Expected Attempts and Judge Cost

Stop-on-acceptance induces a random stopping time that determines test-time cost. Fix a stratum $\ell$ and consider sequentially sampled candidates $D_1, D_2, \ldots$ with judge scores $g_\ell(q, D_i)$. Let

$$a_\ell(q) = \Pr_{D \sim \mathcal{P}_{\text{rand}}} (g_\ell(q, D) \geq \tau_\ell)$$

denote the acceptance rate of judge under the proposal distribution (including both true and false positives). Ignoring truncation, the number of trials until acceptance is approximately geometric with mean $1/a_\ell(q)$. With a finite budget $K_\ell$, the expected number of judge evaluations is

$$\mathbb{E}[\#\text{trials}] = \sum_{i=1}^{K_\ell} \Pr(\text{not accepted in the first } i - 1 \text{ trials}),$$

which is upper bounded by $K_\ell$ and decreases as $a_\ell(q)$ increases. This perspective clarifies how increasing $K_\ell$ improves coverage but raises worst-case cost, while increasing the acceptance threshold $\tau_\ell$ typically lowers $a_\ell(q)$ and increases expected attempts.

## B. Hyperparameter Settings and Training Details

This section summarizes the key hyperparameters used for difficulty stratification and inference; a compact summary is given in Table 7.

**Difficulty stratification.** We estimate $\hat{\rho}(q)$ from $T$ random-demo trials per training query and map $\hat{\rho}(q)$ to difficulty levels using fixed thresholds $(\alpha, \beta, \gamma)$.

**Inference.** For routed queries with $\hat{\ell} \in \{l_1, l_2, l_3\}$, we run stop-on-acceptance with a level-specific judge, acceptance threshold $\tau_{\hat{\ell}}$, and sampling budget $K_{\hat{\ell}}$. We tune $\tau_{l_1}, \tau_{l_2}, \tau_{l_3}$ on a validation set, and use fixed default budgets unless stated otherwise.

Unless stated otherwise, we use the same default inference budgets and acceptance thresholds for all datasets under a fixed backbone.

**Router and judge training procedure.** We train the router and judges using the same offline random-demo trials used for difficulty stratification. Concretely: (i) for each training query $q$, sample $T$ class-balanced demo contexts $D_1, \ldots, D_T \sim \mathcal{P}_{\text{rand}}$ and label each trial with $s(q, D_t)$ using the target LLM; (ii) compute $\hat{\rho}(q) = \frac{1}{T} \sum_{t=1}^{T} s(q, D_t)$ and assign $\ell(q) \in \{l_1, l_2, l_3, l_x\}$ via thresholds $(\alpha, \beta, \gamma)$; (iii) form $\mathcal{D}_{\text{route}} = \{(q, \ell(q))\}$ and $\mathcal{D}_\ell = \{((q, D_t), s(q, D_t)) : \ell(q) = \ell\}$ for $\ell \in \{l_1, l_2, l_3\}$; and (iv) train each judge as a binary classifier on $\mathcal{D}_\ell$. Although the router predicts a 4-way label, we train it using three binary (ordinal) objectives with a shared encoder: $h_1$ distinguishes $l_1$ vs. $\{l_2, l_3, l_x\}$, $h_2$ distinguishes $\{l_1, l_2\}$ vs. $\{l_3, l_x\}$, and $h_3$ distinguishes $\{l_1, l_2, l_3\}$ vs. $\{l_x\}$. At test time, we reconstruct $\hat{\ell}$ by applying these classifiers in order (e.g., predict $l_x$ if $h_3$ rejects, otherwise predict $l_3$ if $h_2$ rejects, otherwise predict $l_2$ if $h_1$ rejects, else $l_1$); we find this hierarchical training improves routing performance. At inference, we tune $\tau_{l_1}, \tau_{l_2}, \tau_{l_3}$ on a validation set and apply stop-on-acceptance with $(K_\ell, \tau_\ell)$.

*Table 8.* Router micro-accuracy (oracle strata) for each dataset. Bold indicates the higher micro-accuracy between the two backbones.

| Dataset | LLaMA3-8B | Qwen2.5-7B |
|---------|-----------|------------|
| TREC    | 77.2      | 80.5       |
| SST-2   | 88.7      | 86.8       |
| SST-5   | 53.8      | 48.5       |
| AGNEWS  | 73.5      | 79.5       |
| MNLI    | 58.5      | 66.2       |

*Table 9.* Full timing breakdown (formatted as `XmYYs`) across datasets.

| Method | Metric | TREC | AGNews | SST-2 | SST-5 | MNLI |
|--------|--------|------|--------|-------|-------|------|
| Uprise | Train time | 13m21s | 16m16s | 9m28s | 14m26s | 13m39s |
| Uprise | Test time | 0m24s | 0m29s | 0m16s | 0m24s | 0m24s |
| Se$^2$ | Train time | 110m29s | 127m02s | 96m36s | 121m21s | 118m47s |
| Se$^2$ | Test time | 0m34s | 0m38s | 0m19s | 0m38s | 0m36s |
| SeDPO | Train time | 130m12s | 177m04s | 150m27s | 156m16s | 171m27s |
| SeDPO | Test time | 0m29s | 0m35s | 0m22s | 0m35s | 0m44s |
| DiSP | Train time | 6m52s | 6m22s | 5m36s | 6m31s | 8m08s |
| DiSP | Test time | 0m04s | 0m05s | 0m04s | 0m05s | 0m05s |

## B.1. Router Micro-Accuracy by Scenario

We report router micro-accuracy as the fraction of test queries whose predicted difficulty stratum matches the oracle stratum. Table 8 summarizes the per-scenario results.

**Relation to end-task accuracy.** Router micro-accuracy is only *indirectly* related to the final task accuracy. The router does not output the task label; instead it selects an inference *policy* (difficulty stratum), which determines which judge(s) and hyperparameters are used (e.g., stop-on-acceptance $(K_\ell, \tau_\ell)$). Consequently, the final accuracy depends not only on whether the router matches the oracle stratum, but also on (i) the confusion structure (which strata it confuses) of the router and (ii) how robust each stratum-specific policy is when applied to queries originating from other strata. Equivalently, with fixed base model and judges, the final accuracy can be viewed as an oracle-stratum-weighted average of the accuracies achieved under the *routed* policy; routing errors only hurt when they send queries to materially mismatched policies (e.g., under-budgeting truly hard queries or using thresholds that prune too aggressively).

Importantly, a *low* router micro-accuracy does not necessarily imply a *low* final task accuracy. First, micro-accuracy is a strict metric: many "errors" are near stratum boundaries, where adjacent strata often induce very similar inference policies, making such misroutes largely benign. Second, the downstream decision procedure is tolerant to mild policy mismatch: stop-on-acceptance can still recover by rejecting early candidates and evaluating later ones. Third, the mapping from stratum to policy is typically designed to be safe—e.g., allocating *more* budget or using slightly more permissive thresholds rarely hurts accuracy as much as it increases cost, so some misroutes mainly affect efficiency rather than correctness. Therefore, the router primarily impacts final accuracy when misrouting is systematic and strongly mismatched (e.g., hard→easy causing consistent under-budgeting or overly strict pruning); otherwise, the dominant accuracy bottleneck may lie in judge quality or the base model itself.

## B.2. Baseline Timing Breakdown

Table 9 provides the per-dataset wall-clock cost for baseline demonstration selection methods used in our comparisons, as well as the end-to-end time of our DiSP pipeline (mean over LLaMA3-8B and Qwen2.5-7B). Train time sums the *score* and *train* stages (offline). Test time sums the *retrieve* stage and the target-LLM inference time.

## C. Representation Probe Results

This appendix reports the complete results for the last-layer MLP probes introduced in Section 4.3. We report AU-ROC/AUPRC for success prediction on prompt instances $x = (q, D)$ across all datasets and target LLM backbones used in

the experiments in Figures 4 to 8, which visualize the full ROC and PR curves for each dataset/backbone setting. Each plot shows one-vs-rest curves for each label, together with micro-/macro-averaged curves; the diagonal dashed line in ROC corresponds to random guessing and the horizontal dashed line in PR indicates the positive-class prevalence.

## D. Prompt Templates and Label Verbalizers

We provide the prompt templates and label verbalizers for each dataset used in our experiments.

### D.1. General Format

We follow a repeated-instruction format. Each example (a labeled demonstration or the unlabeled query) appears as a *block* that contains (i) the dataset-specific instruction, (ii) the input fields, and (iii) a label line that begins with a dataset-specific key (e.g., `Answer:`, `Category:`, or `Relationship:`). For labeled blocks, we place the gold label verbalizer after the key; for the query block, we leave the label blank and score the fixed set of label options. A $k$-shot context concatenates $k$ labeled blocks (one example per class, in a random order) followed by the query block, which ends with the label key and an empty value. At evaluation time, the model is scored over the label verbalizers listed below and we take the most likely label string.

### D.2. TREC (6-way question classification)

**Label verbalizers:** `Abbreviation`, `Entity`, `Description`, `Human`, `Location`, `Numeric`.

```
You are an expert question classifier. Your task is to carefully classify
the given question into one of the following six categories:
Abbreviation, Entity, Description, Human, Location, or Numeric.

Question: <QUESTION>
Category:
```

### D.3. SST-2 (binary sentiment classification)

**Label verbalizers:** `Negative`, `Positive`.

```
You are an expert sentiment analyst. Classify the sentiment of the following
sentence carefully into one of these two categories: negative or positive.

Sentence: <SENTENCE>
Answer:
```

### D.4. SST-5 (5-way sentiment classification)

**Label verbalizers:** `very negative`, `negative`, `neutral`, `positive`, `very positive`.

```
You are an expert sentiment analyst.Classify the sentiment of the following
sentence carefully into one of these five categories:
very negative, negative, neutral, positive, or very positive.

Sentence: <SENTENCE>
Answer:
```

### D.5. AGNEWS (4-way topic classification)

**Label verbalizers:** `World`, `Sports`, `Business`, `Science/Technology`.

```
You are an expert news classifier. Your task is to carefully classify the given
news article into one of the following four categories:
World, Sports, Business, or Science/Technology.
```

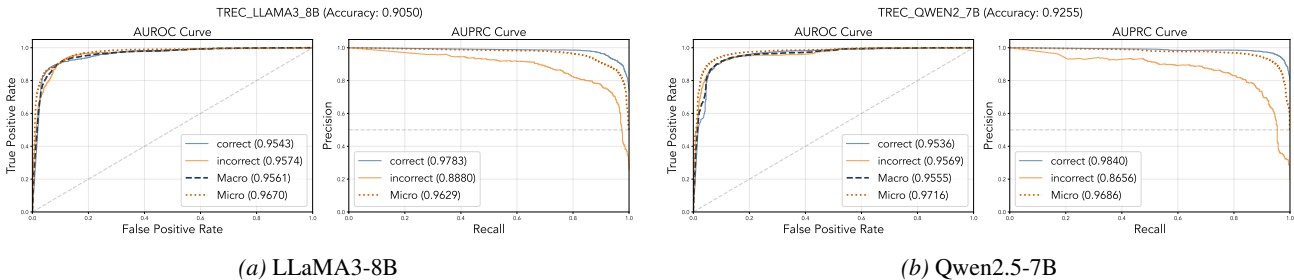

*(a)* LLaMA3-8B

*(b)* Qwen2.5-7B

*Figure 4.* Last-layer MLP probe ROC/PR curves for success prediction on TREC.

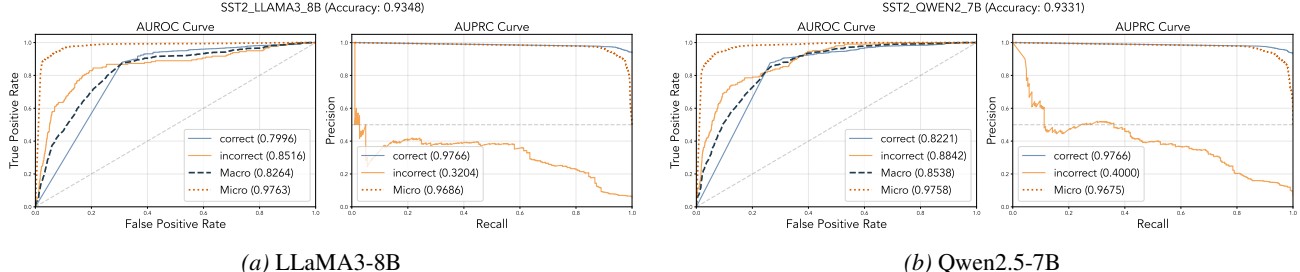

*(a)* LLaMA3-8B

*(b)* Qwen2.5-7B

*Figure 5.* Last-layer MLP probe ROC/PR curves for success prediction on SST-2.

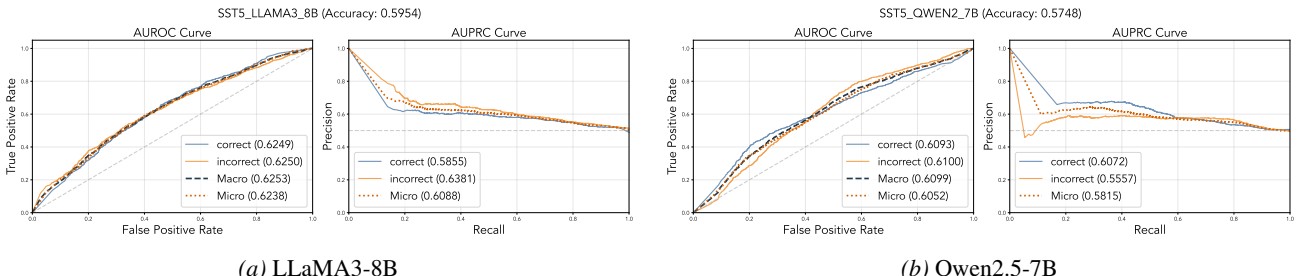

*(a)* LLaMA3-8B

*(b)* Qwen2.5-7B

*Figure 6.* Last-layer MLP probe ROC/PR curves for success prediction on SST-5.

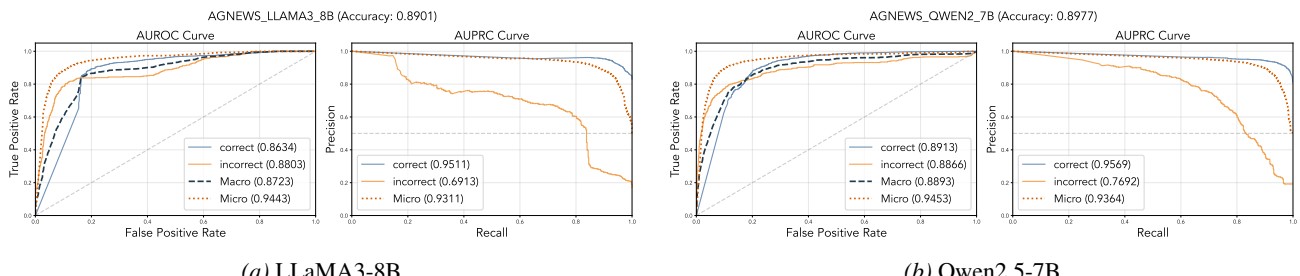

*(a)* LLaMA3-8B

*(b)* Qwen2.5-7B

*Figure 7.* Last-layer MLP probe ROC/PR curves for success prediction on AGNEWS.

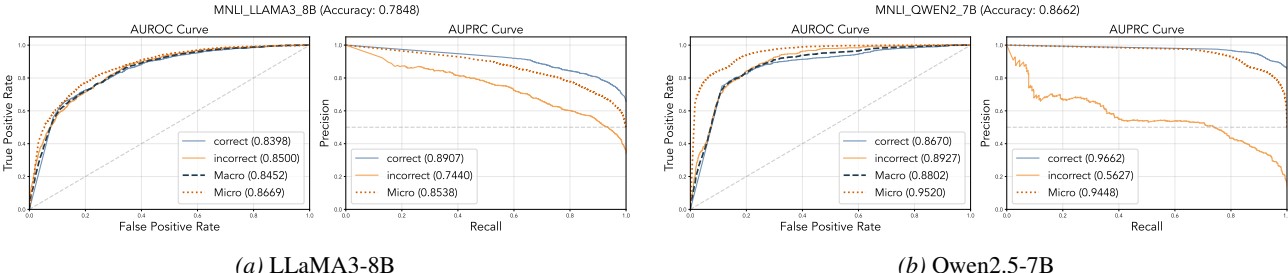

*(a)* LLaMA3-8B

*(b)* Qwen2.5-7B

*Figure 8.* Last-layer MLP probe ROC/PR curves for success prediction on MNLI.

```
Article: <ARTICLE>
Category:
```

### D.6. MNLI (3-way natural language inference)

**Label verbalizers:** `Entailment, Neutral, Contradiction`.

```
You are an expert in natural language inference.
Your task is to carefully classify the relationship between a given
premise and a hypothesis into one of the following three categories:
Entailment, Neutral, or Contradiction.

Premise: <PREMISE>, Hypothesis: <HYPOTHESIS>
Relationship:
```

### D.7. Complete Zero-shot and Few-shot Prompt Examples

We show end-to-end prompts formed by concatenating blocks in our repeated-instruction format.

**Zero-shot (MNLI).**

```
You are an expert in natural language inference.
Your task is to carefully classify the relationship between a given
premise and a hypothesis into one of the following three categories:
Entailment, Neutral, or Contradiction.

Premise: A chef is making dinner,Hypothesis: Someone is cooking.
Relationship:
```

**Few-shot (MNLI, 3-shot).**

```
You are an expert in natural language inference.
Your task is to carefully classify the relationship between a given
premise and a hypothesis into one of the following three categories:
Entailment, Neutral, or Contradiction.

Premise: A man rides a bicycle,Hypothesis: A person rides a bike.
Relationship: Entailment

You are an expert in natural language inference.
Your task is to carefully classify the relationship between a given
premise and a hypothesis into one of the following three categories:
Entailment, Neutral, or Contradiction.

Premise: A woman reads a book,Hypothesis: The woman is drinking coffee.
Relationship: Neutral

You are an expert in natural language inference.
Your task is to carefully classify the relationship between a given
premise and a hypothesis into one of the following three categories:
Entailment, Neutral, or Contradiction.

Premise: Two dogs play outside,Hypothesis: No animals are outdoors.
Relationship: Contradiction
```

You are an expert in natural language inference.
Your task is to carefully classify the relationship between a given
premise and a hypothesis into one of the following three categories:
Entailment, Neutral, or Contradiction.

Premise: A chef is making dinner,Hypothesis: Someone is cooking.
Relationship:

