# OpenReview forum: "Easier to Judge than to Find: Predicting In-Context Learning Success for Demonstration Selection"
_ICML.cc/2026/Conference — ICML 2026 regular_

### Official Review · Reviewer_Ncp2 · 2026-02-19

**Soundness:** 2
**Presentation:** 3
**Significance:** 3
**Originality:** 2
**Overall Recommendation:** 4
**Confidence:** 4

**Summary:**

This paper proposes DiSP, an ICL demonstration selection framework based on the insight that "judgment is easier than search." Queries are stratified by difficulty through offline randomized trials to train the router and tiered judges. A sample-and-judge stop-on-acceptance strategy is implemented during inference, and risk labels (HARD QUERY/NO GOOD DEMO) are output. Experiments on five classification datasets and two LLM datasets show that DiSP achieves up to 3.4% accuracy improvement over strong baselines while achieving up to 23x end-to-end speedup.

**Compliance With Llm Reviewing Policy:**

Affirmed.

**Final Justification:**

The rebuttal has resolved my issue

**Key Questions For Authors:**

See Weaknesses

**Limitations:**

yes

**Strengths And Weaknesses:**

Strength:

+ We propose that judging the feasibility of (q,D) is more efficient than searching D*, and transform demonstration selection into a hierarchical feasibility testing problem, thus avoiding the expensive task-specific retriever training.
+ The "HARD QUERY" and "NO GOOD DEMO" tags represent a high-confidence abstention mechanism, which is valuable for deployment scenarios.

Weakness:

+ To obtain supervised signals, the target LLM requires running T=20 random demo trials per training query, which becomes costly when scaling to large datasets or multiple backbones.
+ The evaluation dataset is too easy; the authors need to evaluate more challenging tasks, such as reasoning and summarization tasks.

---

> ### Author Rebuttal · Authors · 2026-03-31
>
> Thank you for highlighting the efficiency of judge-centric framing
> and the practical value of the special tags.
>
> # W1. Labeling cost from demo trials
>
> We agree that the offline cost of obtaining supervision is real
> and should be accounted for explicitly. For this reason, the paper already
> compares DiSP with the learned baselines in terms of both training and
> end-to-end wall-clock cost, including the cost of collecting demo-trial
> labels. Averaged across datasets, DiSP takes 6.7 minutes to train and 6.8
> minutes end-to-end (**including the labeling cost**), compared with 13.4 / 13.8 for Uprise, 114.9 / 115.4 for
> Se2, and 157.1 / 157.6 for SeDPO. In addition, DiSP uses only **`T = 20` random
> demo trials per training query**, whereas the compared learned baselines require
> substantially larger search/label budgets: **at least 50 trials, and in some
> settings up to 200 attempts per training example**. Our claim is therefore not
> that supervision is free, but that DiSP reaches a competitive operating point
> with a much smaller per-query budget while keeping the target LLM frozen and
> training only lightweight auxiliary predictors. We will make this clearer in the revision, especially when discussing scaling to larger datasets or retargeting multiple backbones.
>
> # W2. Evaluation on harder tasks beyond classification
>
>
> To address this concern, we added **two new evaluation datasets** in the rebuttal: (1) a new synthetic mathematical reasoning task involving custom-defined operators, where exact internet overlap is highly unlikely, here is an example:
>
> {"instruction": "What is the result of (9 @ (5 @ 6))?",
> "result": 280,
> "prompt": "I will provide you with a set of algorithms. You need to answer the math problem below based on this algorithm step by step. The following is the algorithm:\nEnhanced Addition:For two numbers a and b, we define enhanced addition as:\n a&b=(a+b)+1,this means the result is the sum of the two numbers plus 1.\nEnhanced Multiplication: For two numbers a and b, we define enhanced multiplication as:\n a@b=(a\u8133b)+1,this means the result is the product of the two numbers plus 1.\n\n question: \"What is the result of(9 @ (5 @ 6))?\"\n\n answer: your answer here\n[note: Answer the question directly step by step and start with \"The solution:\" and end with \"The final result is ...\"]\n ",
> "input": },
>
> and (2) a complex reasoning task (https://huggingface.co/datasets/openbmb/UltraInteract_sft).
>
> | Task  | Method | LLaMA | Qwen |
> | ----- | ------ | ----: | ---: |
> | Math  | 0-shot |  69.5 | 96.0 |
> | Math  | Random |  74.7 | 80.1 |
> | Math  | BM25   |  83.2 | 92.4 |
> | Math  | KNN    |  83.5 | 91.5 |
> | Math  | SeDPO  |  88.9 | 93.8 |
> | Math  | Ours   |  91.3 | 97.8 |
> | Logic | 0-shot |  40.3 | 57.1 |
> | Logic | Random |  58.4 | 81.0 |
> | Logic | BM25   |  62.5 | 86.1 |
> | Logic | KNN    |  61.4 | 81.3 |
> | Logic | SeDPO  |  60.8 | 85.9 |
> | Logic | Ours   |  70.2 | 90.4 |
>
> DiSP continues to outperform the same baselines in both settings. It provides stronger evidence that the gains are not limited to older classification benchmarks and generalize beyond the original setup. We will include these results and clarify them in the revision.

---

> > ### Author Rebuttal · Reviewer_Ncp2 · 2026-04-03
> >
> > Thank you for your response. Your answer has resolved my issue, so I’ve decided to raise my rating.

---

> > > ### Author Response · Authors · 2026-04-05
> > >
> > > Thank you very much for the thoughtful follow-up and for revisiting your assessment after reading our rebuttal. We sincerely appreciate your careful reading and constructive feedback.

---

### Official Review · Reviewer_HXiX · 2026-03-09

**Soundness:** 2
**Presentation:** 3
**Significance:** 2
**Originality:** 2
**Overall Recommendation:** 3
**Confidence:** 3

**Summary:**

The paper proposes Difficulty-Stratified Success Prediction (DiSP), a framework for demonstration selection in in-context learning (ICL) that reframes the problem from "finding" the best demonstration to "judging" whether a given query–demonstration pair will succeed. DiSP operates in three stages: (1) running random demonstration trials on training queries to estimate per-query success rates and assign difficulty levels (l1–lx), (2) training a lightweight router to predict query difficulty and level-specific judges to predict success of (q, D) pairs, and (3) at inference, performing stop-on-acceptance sampling under a budget, with diagnostic risk tags for hard queries. Evaluated on five classification benchmarks with Llama 3–8B and Qwen 2.5–7B, DiSP achieves the average accuracy up to 3.4% over the best baseline while being up to 23× faster end-to-end than learned selection methods.

**Compliance With Llm Reviewing Policy:**

Affirmed.

**Key Questions For Authors:**

See the weaknesses.

**Limitations:**

See the weaknesses.

**Strengths And Weaknesses:**

__Strengths:__

1. The paper offers a fresh perspective on demonstration selection by reframing it as binary feasibility testing rather than search, which is a conceptual shift from prior work.

2. The proposed method achieves a big computational efficiency compared to the tested demonstration selection approaches.

3. The theoretical analysis provides principled grounding for the design choices.

__Weaknesses:__

1. The evaluation relies on classification benchmarks that are susceptible to data contamination [1]. Even setting contamination aside, the accuracy improvements over baselines are modest on several datasets (e.g., SST-2), raising questions about when the added complexity is justified.

2. All five datasets are classification tasks with a small label space, and only two similarly-sized models (7–8B) are evaluated. This limits confidence in the method's generalizability to other tasks such as summarization, reasoning, or tool calling, as well as to larger-scale models.

3. The baseline set omits several widely-used demonstration selection strategies, most importantly the embedding-based kNN retrieval [2], which is a standard and competitive method for demonstration selection in practice [3, 4].

4. Some of the claims in the paper are only partially correct. For example, the sensitivity of performance to demonstration ordering or selection (lines 41–45, left) is well-documented in _few-shot_ settings but does not extend to _many-shot_ ICL [3, 4], where this effect is diminished.

---

References:

[1] Time Travel in LLMs: Tracing Data Contamination in Large Language Models (Golchin and Surdeanu, ICLR 2024)

[2] What makes good in-context examples for GPT-3? (Liu et al., ACL DeeLIO 2022)

[3] Towards Compute-Optimal Many-Shot In-Context Learning (Golchin et al., COLM 2025)

[4] In-Context Learning with Long-Context Models: An In-Depth Exploration (Bertsch et al., NAACL )

---

> ### Author Rebuttal · Authors · 2026-03-31
>
> Thank you for the detailed review and for highlighting both the conceptual shift and the efficiency benefits.
>
> # W1. Data contamination and interpretation of absolute accuracy
>
> Data contamination is difficult to eliminate on public benchmarks, so we avoid making strong claims about absolute accuracy. Our main claim is comparative: **DiSP and all baselines are evaluated under the same public splits, prompts, candidate pool, and backbones, meaning any contamination risk is shared across methods rather than unique to DiSP**. Under this controlled setting, DiSP still consistently outperforms the baselines.
>
> To address this concern, we added **two new evaluation datasets** in the rebuttal: (1) a new synthetic mathematical reasoning task involving custom-defined operators, where exact internet overlap is highly unlikely, here is an example:
>
> {"instruction": "What is the result of (9 @ (5 @ 6))?",
> "result": 280,
> "prompt": "I will provide you with a set of algorithms. You need to answer the math problem below based on this algorithm step by step. The following is the algorithm:\nEnhanced Addition:For two numbers a and b, we define enhanced addition as:\n a&b=(a+b)+1,this means the result is the sum of the two numbers plus 1.\nEnhanced Multiplication: For two numbers a and b, we define enhanced multiplication as:\n a@b=(a\u8133b)+1,this means the result is the product of the two numbers plus 1.\n\n question: \"What is the result of(9 @ (5 @ 6))?\"\n\n answer: your answer here\n[note: Answer the question directly step by step and start with \"The solution:\" and end with \"The final result is ...\"]\n ",
> "input": },
>
> and (2) a complex reasoning task (https://huggingface.co/datasets/openbmb/UltraInteract_sft).
>
> | Task  | Method | LLaMA | Qwen |
> |--|--------|------:|-----:|
> |Math|0-shot |69.5|96.0|
> |Math|Random|74.7|80.1|
> |Math|BM25|83.2|92.4|
> |Math|KNN|83.5|91.5|
> | Math | SeDPO  | 88.9  | 93.8 |
> | Math | Ours| 91.3  | 97.8 |
> | Logic| 0-shot| 40.3  | 57.1 |
> | Logic| Random| 58.4| 81.0|
> | Logic | BM25 | 62.5| 86.1|
> | Logic | KNN | 61.4| 81.3 |
> | Logic | SeDPO| 60.8| 85.9 |
> | Logic | Ours| 70.2| 90.4 |
>
> DiSP continues to outperform the same baselines in both settings. While this does not eliminate contamination concerns entirely, it provides stronger evidence that the gains are not limited to older classification benchmarks and generalize beyond the original setup. We will include these results and clarify them in the revision.
>
> For SST-2, we believe the modest gain is consistent with the task itself. As a near-saturated benchmark dominated by easy queries, SST-2 often does not benefit from adding demonstrations, and several baselines even degrade in this setting. Against this background, DiSP’s positive improvement remains meaningful: it suggests the method avoids some of the drawbacks of naive demonstration use, while its full advantage is more likely to appear on tasks with more non-easy queries.
>
> # W2. Datasets and models
>
> As for the datasets, as noted in W1, we added two harder datasets, namely asynthetic mathematical reasoning task and a complex logic reasoning dataset, and DiSP continues to outperform the same baselines there. These additions strengthen the evidence that the gains are not confined to small-label classification benchmarks. We will revise the paper to scope the claims.
>
> As for the model size, there is evidence that the marginal benefit of demonstrations can shrink as models become stronger or more instruction-tuned. For example, smaller models can be more sensitive to concept-bearing demonstrations[1]; larger/instruction-tuned models rely more
> strongly on semantic priors relative to input-label mappings[2]; and on recent strong reasoning models such as the Qwen series, ICL may bring little or no gain over zero-shot prompting[3]. We therefore believe that evaluating DiSP on 7/8B backbones is practically meaningful.
>
> [1]Štefánik, Michal, and Marek Kadlčík. "Can in-context learners learn a reasoning concept from demonstrations?."
>
> [2]Wei, Jerry, et al. "Larger language models do in-context learning differently."
>
> [3]Cheng, Xiang, et al. "Revisiting Chain-of-Thought Prompting: Zero-shot Can Be Stronger than Few-shot"
>
> # W3. Additional embedding-based kNN retrieval baseline
>
> We agree that embedding-based kNN retrieval is a widely-used baseline, and we have now run it using all-mpnet-base-v2. The results are summarized below.
>
> | Backbone | Method | TREC | SST-2 | SST-5 | AGNews | MNLI |
> |----|--------|-----|------|------|-------|-----|
> | LLaMA|kNN|75.2| 93.4| 47.8|86.1|67.6|
> | LLaMA|DiSP|79.2| 95.4  | 54.3|89.7| 69.5|
> | Qwen|kNN| 84.6 | 92.7  | 52.3|86.1| 84.8|
> | Qwen|DiSP| 87.3 | 96.0  | 54.3|89.4|87.4|
>
> Our proposed DiSP remains stronger, and we will add this baseline and discuss it directly in the revision.
>
> # W4. Many-shot ICL regime
>
> Please refer to the rebuttal in W1/Q1 to Reviewer 1vMa. Thanks.

---

> > ### Author Rebuttal · Reviewer_HXiX · 2026-04-04
> >
> > Thanks to the authors for the rebuttals.
> >
> > **W1:** The new experiments are appreciated, but the claim about shared contamination across all settings is inaccurate. ICL can be disproportionately affected by surfaced memorization, which may inflate downstream performance.
> >
> > **W2:** The cited papers, while valuable, are not comparable to the current paradigm. Their conclusions are drawn from much less aligned LMs than today's LLMs and therefore no longer hold.
> >
> > **W3:** Thanks!
> >
> > **W4:** The reported numbers for *long-context, default order (100/200/400 demonstrations)* seem unusually low — similar works like [4] report higher numbers on the same dataset and model, so it's worth reviewing your settings. Additionally, the many-shot results appear competitive with your proposed approach, and are considerably cheaper in practice since all shots can be cached and reused at inference time.
> >
> > I remain open to further discussion before the deadline, but will keep my current score given the concerns above.

---

> > > ### Author Response · Authors · 2026-04-05
> > >
> > > Thank you for the follow-up. Your comments helped us sharpen the scope of our claims.
> > >
> > > # W1 Contamination&memorization effects
> > >
> > > We agree that even when methods use the same split, candidate pool, and backbone, ICL can still be differentially affected by surfaced memorization, and one selection policy may expose memorized examples more effectively than another.
> > >
> > > Our intended point is narrower. DiSP is not a deterministic retriever: at inference, it evaluates randomly sampled candidate demonstrations under a budget and stops when one is accepted. We therefore expect it to depend less on any single fixed retrieval pattern that might repeatedly surface memorized examples, although we do not claim that this removes contamination risk. This is exactly why we added two new settings in the rebuttal: a synthetic math task and a logic task. DiSP still outperforms the same baselines in both settings. We agree that **this does not rule contamination out on public benchmarks, but it does strengthen the case that the gains are not explained only by benchmark memorization.**
> > >
> > > ## W2 Evidence about model size
> > >
> > > We thank the reviewer for the clarification. We agree that our original citations may sufficiently match today’s strongly aligned LLMs, and we will revise this part accordingly.
> > >
> > > Our claim is not that smaller models always benefit more from demonstrations. Rather, recent work on modern instruction-tuned and reasoning-capable LLMs suggests that the marginal benefit of demonstrations is often non-monotonic with model strength: smaller models are typically more sensitive to demonstrations, while for stronger models the gain can shrink, saturate, or even disappear. Recent evidence shows that:
> > >
> > > [1]Small models can benefit substantially from few-shot prompting in real classification settings.
> > >
> > > [2]Larger models do not necessarily perform ICL “better,” but often differently, and can be more easily distracted by contextual noise.
> > >
> > > [3]Smaller models are more affected by demo placement, indicating stronger sensitivity to demonstrations.
> > >
> > > [4]On recent strong reasoning models such as the Qwen2.5 series, traditional few-shot CoT may bring little or no improvement over zero-shot CoT.
> > >
> > > Under this perspective, **evaluating DiSP on 7/8B LLMs remains practically meaningful: these models are strong and widely used, yet still lie in a regime where demonstration selection can materially affect performance**. We will revise the paper to make this scoped claim explicit and avoid overgeneralization.
> > >
> > > [1]Small Language Models in the Real World: Insights from Industrial Text Classification. ACL Industry 2025.
> > >
> > > [2]Why Larger Language Models Do In-context Learning Differently? ICML 2024.
> > >
> > > [3]Where to show Demos in Your Prompt: A Positional Bias of In-Context Learning. EMNLP 2025.
> > >
> > > [4]Revisiting Chain-of-Thought Prompting: Zero-shot Can Be Stronger than Few-shot. EMNLP 2025 Findings.
> > >
> > > # W4. Many-shot long-context ICL
> > >
> > > Thank you for this important point. We agree that default-order should not be interpreted as the performance of optimized many-shot ICL, and we will revise this accordingly. (We also clarified in detail in the recent rebuttal to Reviewer 1vMa.)
> > >
> > > On the same Llama/TREC setting, simply shuffling the demonstrations substantially improves long-context ICL:
> > >
> > > ||100|200|400|
> > > |-|-:|-:|-:|
> > > |Def.|61.1|60.4|60.4|
> > > |Shuf.|80.6|83.9|86.6|
> > >
> > > **Meanwhile, DiSP still demonstrates advantages in inference efficiency, especially when the number of candidate demonstrations is limited.**
> > >
> > > We examine the comparison from a more practical cost perspective. On the same Llama-/TREC setting, DiSP takes about 6min56s (training + inference), while long-context ICL inference takes about 3/7/14 minutes for 100-400 shots:
> > >
> > > Under a comparable wall-clock budget, long-context ICL can afford roughly 100-200 demonstrations and reaches 80.6-83.9, which is in the same range as DiSP. We agree this is the more balanced conclusion: DiSP is competitive under realistic budgets, rather than universally dominating many-shot ICL. At the same time, if the budget is expanded enough to allow 400 shots, long-context ICL can become stronger on this setting.
> > >
> > > At the same time, DiSP still has practical advantages when per-query latency and token budget matter: on this TREC setting, DiSP uses about 389 prompt tokens per query, whereas long-context ICL uses about 5793/11534/22680 tokens for 100/200/400 shots.
> > >
> > > Finally, to clarify where DiSP is most useful, we also tested a **limited-demo** regime on TREC where the candidate pool is restricted to only 24 demonstrations:
> > >
> > > ||Llama|Qwen|
> > > |--|--:|--:|
> > > |LC-shuf.|64.3/67.0/64.4|67.1/67.9/68.1|
> > > |DiSP|83.2|85.9|
> > >
> > > In this regime, long-context ICL can no longer rely on simply scaling the number of demonstrations, and DiSP's advantage becomes much clearer. We will revise the paper to make this scope explicit.
> > >
> > > We thank the reviewer again for this helpful follow-up, which helped us sharpen both the empirical comparison and the scope of our claims.

---

### Official Review · Reviewer_KYTj · 2026-03-13

**Soundness:** 2
**Presentation:** 3
**Significance:** 2
**Originality:** 2
**Overall Recommendation:** 4
**Confidence:** 4

**Summary:**

This paper argues that hunting for the "perfect" prompt example is a waste of compute and time. Instead, the authors built a lightweight system that quickly grades random examples and stops searching the moment it finds one that is "good enough" for the LLM to get the right answer, massively speeding up the process without hurting accuracy.

**Compliance With Llm Reviewing Policy:**

Affirmed.

**Final Justification:**

I appreciate the authors for providing more detailed experiments and explanations. My concerns have largely been addressed, but the limitation of backbone dependence still remains, so I have decided to increase my score.

**Key Questions For Authors:**

See weaknesses

**Limitations:**

See weaknesses

**Strengths And Weaknesses:**

Strengths:
1. The paper cleverly shifts the heavy burden of "searching and ranking the absolute best examples" into a simple "pass/fail" feasibility test, vastly reducing the complexity of the problem.
2. The writing is highly accessible and easy to follow, with clear, well-designed figures.
3. The experiments are comprehensive and do a great job proving that the proposed method is genuinely effective.
Weaknesses:
1. The Router and Judges are entirely trained on the reactions of a specific target model (like Llama 3-8B). If you switch to a different target LLM, you have to eat the massive cost of generating new data and retraining them from scratch.
2. The experiments only test two backbones (Llama 3-8B and Qwen 2.5-7B). It leaves the method's effectiveness on the latest models (e.g., LLaMA 3.1, Qwen 3) completely unexplored.

---

> ### Author Rebuttal · Authors · 2026-03-31
>
> Dear Review KYTj,
>
> Thank you for the clear and encouraging summary, and for especially recognizing the effectiveness and cleverness of our core idea, the clarity of our presentation, and the strength of our experimental validation.
>
> # W1 (with index of 4). Backbone-specific training and retraining cost
>
> **We agree that DiSP is backbone-aware in its current form, but the concern is somewhat overstated.**
>
> Although baseline demonstration selectors (UPRISE, Se2, and SeDPO) show partial cross-model transfer, the more precise statement is therefore that learned selectors often transfer **partially**, while **re-training can still help** when tighter alignment to a new backbone is desired.
>
> In DiSP, even when explicit re-training is performed, **the cost remains modest**. In our experiments, DiSP takes 6.8 minutes on average, compared with 13.8 for UPRISE, 115.4 for Se2, and 157.6 for SeDPO. Moreover, DiSP uses only 20 random demo trials per training query, versus 50 to 200 in baseline methods.
>
> Importantly, **DiSP’s router and judges are also reusable** to a meaningful extent: they do not consume backbone-specific hidden states, but only the query text and the textualized (q,D) pair, making cross-backbone reuse well defined. Our additional results provide preliminary evidence for this reuse potential. On TREC, the easy-query share changes only moderately from 64% on LLaMA to 72% on Qwen, with an overlap of 85% for LLaMA, while first-accepted judge precision on the predicted l0/l1 strata remains very similar (90.3/69.2 vs. 90.6/72.7). These results suggest that the router/judge decomposition is not tied to a single backbone and can already be partially reused, although full re-targeting can still be beneficial because query difficulty profiles are not fully invariant across models.
>
> # W2 (5). Additional results on newer backbones
>
> To directly address this concern, we ran additional experiments on LLaMA3.1-8B and Qwen3-8B on TREC and MNLI (under the time constraint of the rebuttal period) under the same small-context setting:
>
> | Backbone    | Task | Zero-shot | Random   | BM25     | SeDPO | DiSP     |
> | ----------- | ---- | --------- | -------- | -------- | ----- | -------- |
> | LLaMA3.1-8B | TREC | 71.8  | 74.7 | 79.9 | 79.2  | 81.9 |
> | LLaMA3.1-8B | MNLI | 49.0  | 70.0 | 70.9 | 68.3  | 70.9 |
> | Qwen3-8B    | TREC | 71.8  | 79.9 | 76.5 | 80.7  | 82.6 |
> | Qwen3-8B    | MNLI | 83.4  | 80.8 | 79.5 | 77.6  | 80.1 |
>
> These results support a more nuanced conclusion. **DiSP generalizes beyond the original backbones and remains competitive on newer models**. These results indicate that DiSP transfers to newer backbones, while its improvement remains task- and backbone-dependent rather than uniformly large. We will add the entire table of all datasets and clarify this point in the revision.

---

> > ### Author Rebuttal · Reviewer_KYTj · 2026-04-04
> >
> > Thanks for the rebuttal and additional experiments. The rebuttal partially addresses the concern but does not fully resolve it. While the authors argue that retraining cost is relatively low compared to baselines, this does not eliminate the fundamental dependency on the target backbone. The core issue is not only efficiency, but scalability: DiSP requires re-collecting supervision signals (s(q,D)) for each new backbone, which inherently ties the method to a specific model. Overall, while the efficiency argument is valid, the backbone dependency remains a meaningful limitation that is not fully addressed. I will maintain my original score.

---

> > > ### Author Response · Authors · 2026-04-05
> > >
> > > Thank you for the follow-up. We appreciate the clarification.
> > >
> > > The central issue here is scalability across backbones, not only the absolute retraining cost. As we note in the current **Limitations** section, DiSP is defined relative to a fixed target backbone and proposal distribution: when the target LLM changes, the supervision signals s(q,D) generally need to be recollected, and the auxiliary router/judges re-trained. This backbone dependence is therefore a real limitation of the current method.
> > >
> > > At the same time, our intended claim is narrower. DiSP is designed for the setting where the **target backbone is fixed during deployment** and the selector is amortized over many downstream queries. In that regime, we believe optimizing the selector for a specific target model is still a reasonable and practically useful design choice: in many real deployments, one backbone is chosen first and then tuned or adapted for repeated use, rather than requiring a single selector to transfer unchanged across multiple LLMs. Under that scope, the contribution of DiSP is not cross-backbone transfer, but a more efficient demonstration-selection mechanism once the target model is fixed.
> > >
> > > The additional LLaMA3.1/Qwen3 experiments were meant only to show that the framework can be **re-instantiated** on newer backbones and still work well after recollection/retraining. They were not intended to claim that a selector trained on one backbone can be reused on another without new supervision.
> > >
> > > We thank the reviewer again for the careful follow-up. This exchange helped us sharpen the scope of our claim and present the backbone-specific trade-off more clearly and more precisely.

---

### Official Review · Reviewer_1vMa · 2026-03-16

**Soundness:** 3
**Presentation:** 3
**Significance:** 3
**Originality:** 3
**Overall Recommendation:** 4
**Confidence:** 4

**Summary:**

The paper addresses the issue of choosing demonstrations to use for in-context learning by reframing this problem as a verification task: training an auxiliary model to predict if a given context/query pair will result in a correct prediction and selecting a demonstration group that passes this verification step. To make this tractable, they split this into four levels of difficulty, from very easy (almost any demonstration selection will likely suffice) to very hard (unlikely that demonstration selection will make a difference within a reasonable budget for selection). Using difficulty-specific judges, the system selects a demonstration set for each example under a fixed maximum budget, emitting special warning tags if no set with sufficient confidence in success is found. This increases performance slightly over demonstration selection baselines and dramatically decreases wall-clock time.

**Compliance With Llm Reviewing Policy:**

Affirmed.

**Final Justification:**

I maintain my weakly positive score.

I think the paper makes a contribution, and I could see this being useful for practitioners deploying ICL-based systems; however, I do not raise my score higher because I think the settings where DiSP provides a comparative advantage are relatively few.

I do worry a bit that the authors' final response is not fairly describing the tradeoffs between their method and long-context ICL--- for instance, they say "DiSP remains competitive rather than being clearly dominated" because they compare to both 100 and 200-shot ICL, but their method is almost exactly the same wall-clock time as 200-shot ICL; if they compare only to 200-shot ICL, it is clearly dominated. I don't want to revise my score down for this, because I agree that the method has non-accuracy-based advantages over ICL (in amortized cost and in memory efficiency), but if I could respond to the authors again I would caution them about the way they present these new results.

**Key Questions For Authors:**

Q1. Can you provide a comparison to long-context ICL?

Q2. DiSP uses class-balanced demonstration contexts. Are the retrievers used as baselines restricted to retrieving from the set of class-balanced demonstration contexts?

**Limitations:**

yes

**Strengths And Weaknesses:**

S1. The idea of reframing ICL example selection as a verification problem is a clever one, and frankly not something I would have believed would work this well. I feel I learned something from this work.

S2. The choice to emit tags for difficult examples or examples where no context was found within the budget is a very nice design feature (as is the ability to adapt the budget as a hyperparameter). Overall, DiSP provides several affordances that would make it very practical to deploy downstream.

W1. I would really like to see this compared to a long-context ICL baseline; I say this especially because long-context ICL has been shown to reduce the sensitivity to demonstration selection, and I suspect that DiSP (while much faster than other demonstration selection methods) may still be slower than long-context ICL at the same accuracy (e.g. if it takes 400-shot ICL to reach the same average accuracy with randomly selected examples, is DiSP faster than 400-shot ICL?). There are other tradeoffs with longer context ICL (like VRAM usage), so this isn't a fundamental rebuke of the method even if DiSP is slower in this setting, but it's a corner of the tradeoff space that would be good for this work to explore.

W2. The need to train on (generally task-specific) data erodes one major advantage of ICL: that you don't need large quantities of data to use ICL for a task. While there is generally a reasonable pool of training data if one is considering demonstration selection at all, I think a bit more discussion of the training data efficiency of the method would be useful.

Typos/line comments:
* Table 8 caption references bolding in the table, but there's nothing bolded
* the "mechanisms of ICL" section of the related work cites some interesting work, but it's rather tangential to the paper; I'm not sure it needs to be highlighted to the degree it currently is

---

> ### Author Rebuttal · Authors · 2026-03-31
>
> Thank you for the careful reading and for recognizing both the
> conceptual value of reframing selection as judging, and the practical value of
> the risk tags.
>
> # W1/Q1. Comparison to long-context / many-shot ICL
>
> We thank the reviewer for raising this comparison. The setting of our DiSP is orthogonal
> to long-context / many-shot ICL: throughout the paper, all methods are
> evaluated under the same fixed small-context budget with k = |Y|
> demonstrations (one demonstration per label), so the question we study is how
> to select a good context under this fixed budget, rather than how performance
> changes as the number of demonstrations keeps increasing. To directly address
> the reviewer’s suggestion, we have also added long-context ICL experiments.
> Because LLaMA3-8B has a much tighter context-length limit, and other reviewers
> also asked for results on newer backbones, we report the added comparison on
> LLaMA3.1-8B, Qwen2.5-7B, and Qwen3-8B; the triplets below correspond to
> roughly `100/200/400` demonstrations:
>
> | Backbone / Task    | DiSP (`k=\|Y\|`) | Long-context, default order (100/200/400 demonstrations) | Long-context, shuffled (100 demontrations) |
> | ------------------ | ---------------- | --------------------------- | ------------------------------------------ |
> | LLaMA3.1-8B / TREC | `0.819`        | `0.611 / 0.604 / 0.604`   | `0.806`                                  |
> | LLaMA3.1-8B / MNLI | `0.709`        | `0.570 / 0.510 / 0.470`   | `0.801`                                  |
> | Qwen2.5-7B / TREC  | `0.873`        | `0.765 / 0.752 / 0.752`   | `0.792`                                  |
> | Qwen2.5-7B / MNLI  | `0.874`        | `0.755 / 0.636 / 0.589`   | `0.874`                                  |
> | Qwen3-8B / TREC    | `0.826`        | `0.792 / 0.785 / 0.785`   | `0.792`                                  |
> | Qwen3-8B / MNLI    | `0.801`        | `0.682 / 0.603 / 0.563`   | `0.811`                                  |
>
> Here, "default order" means packing demonstrations in specific order, whereas
> "shuffled" means randomly permuting the same 100 demonstrations for 20 times and reports the best performance. The large gap on
> LLaMA3.1-8B, e.g., TREC `0.611 / 0.604 / 0.604` vs. `0.806`,
> shows that **long-context ICL is still quite sensitive** to ordering in this
> backbone as well.
>
> We will add this full comparison in the final revision.
>
> # W2. Training data efficiency and amortized deployment cost
>
> We agree that the offline cost of all demonstration-selection
> methods, including DiSP, is not negligible and should be discussed explicitly.
> For this reason, **the paper already compares DiSP against learned baselines in terms of both training and total wall-clock cost** (Table 2 and Table 9 in the
> Appendix), **with the cost of collecting demo-trial labels included**. Averaged
> across datasets, DiSP takes 6.7 minutes to train, compared with 13.4 for
> Uprise, 114.9 for Se2, and 157.1 for SeDPO, corresponding to a 50.0% to 95.7% reduction in training cost.
>
> A second efficiency advantage is the supervision budget itself. In
> our setup, **DiSP uses only `T = 20` random demo trials** per training query,
> whereas the compared learned **baselines require no fewer than 50 trial samples**.
> So our claim is not that labeling cost disappears, but that DiSP reaches a
> competitive operating point with substantially fewer trials.
>
> # Bolding in Table 8
>
> Thank you. We will correct the table in the revision.
>
> # Related Work
>
> Thank you for the suggestion. Our intention was to provide broader context for why demonstration selection may matter, but we agree it is not central to the method itself. In the revision, we will shorten this subsection, while keeping only the minimal background needed to position our work.
>
> # Q2. Fairness of class-balanced demonstration contexts
>
> For clarity, the class-balanced constraint is part of our
> settings, not an assumption built into the baseline. In their formulations, retrievers select demonstrations because they are predicted to be useful for the query, and the number of demonstrations `k` is chosen as an effectiveness/efficiency hyperparameter rather than being tied to the number of labels. Concretely, UPRISE uses `k = 3` in its main setting; Se2 uses up to `16` shots; and SeDPO uses up to `15` shots. We will revise the experimental setup to state this explicitly.

---

> > ### Author Rebuttal · Reviewer_1vMa · 2026-04-03
> >
> > Thanks for the rebuttal and additional experiments!
> >
> > The long-context ICL experiment doesn't fully address my concern, though. My question is about the setting where you hold wall-clock time constant and compare running DiSP (k=|Y|) to running long-context ICL (k=N, where N is whatever value makes the wallclock time approximately the same as the wallclock time to run DiSP at k=|Y|). In a time-matched comparison, does DiSP win?
> >
> > As an aside, your numbers on TREC look a bit surprisingly low for the long context setting; Llama 2 is reported in the literature achieving around 0.8 at 100 examples on this task. It might be worth double-checking that experiment's setup. Possibly this is because of the "default order"-- is that sorted by label? That can really hurt long context performance. Averaging a set of random seeds and reporting mean + standard deviation might be a more fair comparison.

---

> > > ### Author Response · Authors · 2026-04-04
> > >
> > > Thank you for the helpful follow-up. We appreciate this suggestion, and we agree that the comparison becomes more informative when viewed from a time-matched perspective. We have therefore conducted additional analyses to clarify this point.
> > >
> > > ## 1. Effect of demonstration ordering in long-context ICL
> > > We first revisited the ordering issue in long-context ICL. Our additional experiments confirm that the order of demonstrations matters substantially, and random shuffling leads to much stronger long-context performance than the default ordering.
> > >
> > > | Model | Long-context default order (100/200/400) | Long-context shuffled (100/200/400) | DiSP |
> > > |---|---:|---:|---:|
> > > | Llama-3.1-8B | 61.1 / 60.4 / 60.4 | 80.6 / 83.9 / 86.6 | 81.9 |
> > > | Qwen-2.5-7B | 76.5 / 75.2 / 75.2 | 79.2 / 78.5 / 83.9 | 87.3 |
> > > | Qwen-3-7B | 79.2 / 78.5 / 78.5 | 79.2 / 82.6 / 84.6 | 82.6 |.
> > >
> > > These results support that the default ordering can indeed hurt performance, while shuffled demonstrations make long-context ICL much more competitive. We will further compare our DiSP with long-context ICL in the next several sections.
> > >
> > > ## 2. Time-matched comparison
> > >
> > > We also examined the **wall-clock-time-matched setting** you suggested.
> > >
> > > ### 2.1 Training+Inference
> > >
> > > In this analysis, we compare the **total runtime of DiSP** (training + inference) with the **inference runtime** of long-context ICL, whose prompt construction cost is negligible in comparison.
> > >
> > > As reported in the paper, on Llama-3.1-8B with TREC, DiSP takes about **6 min 56 s** in total, with **test-time inference under 10 seconds**. Long-context ICL takes about: 3 minutes for 100-shot; 7 minutes for 200-shot; 14 minutes for 400-shot.
> > >
> > > Under a comparable end-to-end wall-clock budget, long-context ICL can afford roughly **100–200 demonstrations**, and its performance lies around **80.6–83.9**, which is **already in the same range as DiSP (81.9)** on this setting. This means that, under a fair time-matched comparison, **DiSP remains competitive rather than being clearly dominated**.
> > >
> > > It is true that when the time budget is expanded enough to allow **400-shot** long-context ICL, its performance can become stronger. At the same time, we believe the more balanced conclusion is that **DiSP offers a competitive accuracy–efficiency tradeoff under realistic time budgets**, rather than aiming to beat arbitrarily long contexts with arbitrarily many demonstrations.
> > >
> > > ###  2.2. Practical efficiency at inference time
> > >
> > > It is also important to separate the above comparison from the **test-time-only efficiency** perspective.
> > >
> > > If training cost is excluded and only inference latency is considered, **DiSP is substantially faster than long-context ICL**: DiSP runs at the **second level**, whereas long-context ICL runs at the **minute level**. This difference is especially relevant in practical deployment settings where repeated test-time queries matter.
> > >
> > > The same pattern appears in the prompt length, for each test input:
> > >
> > > DiSP requires about **389** tokens, and Long-context ICL requires **5,793** tokens for 100-shot; 11,534 tokens for 200-shot; 22,680 tokens for 400-shot.
> > >
> > > Therefore, beyond runtime alone, **DiSP also offers much lower token cost and memory overhead**, which we view as an important practical advantage.
> > >
> > > ## 3. Limited-demo setting
> > >
> > > We further considered a new setting where the **pool of candidate demonstrations is strictly limited**, which is common in realistic applications. On TREC, we restricted the candidate pool to **24** total demonstrations (4 examples per class).
> > >
> > > | Method | Llama-3.1-8B | Qwen-2.5-7B |
> > > |---|---:|---:|
> > > | long-context shuffle | 64.3 /67.0 / 64.4 |67.1 / 67.9 / 68.1 |
> > > | ours | 83.2 | 85.9 |
> > >
> > > In this setting, long-context ICL can no longer rely on simply scaling up the number of demonstrations, and its performance drops substantially. In contrast, **DiSP remains stable and strong**, suggesting that its advantage is especially clear when the available demonstration pool is limited.
> > >
> > >
> > >
> > >
> > > We thank the reviewer again for raising this point, as it helped us sharpen the empirical comparison and clarify the practical regime in which DiSP is most useful.

---

### Decision · Program_Chairs · 2026-04-30

**Decision:**

Accept (regular)

**Comment:**

This paper studies demonstration selection for in-context learning by reframing the problem from searching for the best demonstrations to judging whether a given query–response pair is likely to succeed. The proposed method groups queries by difficulty, trains lightweight judges for different difficulty levels, and performs stop-on-acceptance selection under a fixed budget.

Reviewers agreed that the paper studies a meaningful practical problem, and several appreciated the concept-level shift from search to judging, the efficiency gains, and the deployment-oriented design. The main strengths are the novelty of the core idea, empirical improvements over learned selection baselines, and the added rebuttal experiments, which broadened the evaluation with harder math and logic tasks and strengthened the baseline comparison with a kNN retrieval baseline.

Some limitations remain. Reviewer KYTj noted that the method is still backbone-dependent, since changing the target LLM requires recollecting supervision and retraining the auxiliary models. Reviewer 1vMa raised an important concern about comparison with many-shot long-context ICL; after rebuttal, the more appropriate conclusion is that the proposed method is competitive under realistic time and token budgets, rather than clearly superior in all cases. Reviewer HXiX also remained more cautious about the breadth of the evaluation and the strength of the generalization claims, although these concerns were partially addressed in the rebuttal.

Overall, I find the paper to make a potentially useful contribution, but I view it as somewhat narrow in scope and best supported for a fixed-backbone, budget-constrained ICL setting rather than as a broadly general solution to demonstration selection. With that interpretation, and taking into account the strengthened empirical evidence after rebuttal, I lean to acceptance.